



# The influence of anomalous atmospheric conditions at Ny-Ålesund on clouds and their radiative effect

Tatiana Nomokonova[1], Kerstin Ebell[1], Ulrich Löhnert[1], Marion Maturilli[2], and Christoph Ritter[2]

[1]Institute for Geophysics and Meteorology, University of Cologne, Cologne, Germany
[2]Alfred Wegener Institute Helmholtz Centre for Polar and Marine Research, Potsdam, Germany

**Correspondence:** Tatiana Nomokonova (tnomokon@uni-koeln.de)

**Abstract.** This study analyses occurrence of events with increased and decreased integrated water vapor (IWV) and atmospheric temperature (T) at the Arctic site Ny-Ålesund and their relation to cloud properties and the surface cloud radiative effect (CRE). For this study, we used almost 2.5 years (from June 2016 to October 2018) of ground-based cloud observations processed with the Cloudnet algorithm, IWV and T from microwave radiometer (MWR), long-term radiosonde observations, and backward trajectories FLEXTRA. Moist and dry anomalies were found to be associated with North Atlantic flows and air circulations in the Arctic region, respectively. The amount of water vapor is often correlated with cloud occurrence, presence of cloud liquid water, liquid and ice water path (LWP and IWP). In turn, changes in the cloud properties cause differences in surface CRE. During dry anomalies, in autumn, winter, and spring, the mean net surface CRE was lower by 2–37 W m$^{-2}$ with respect to normal conditions, while in summer the cloud related surface cooling was reduced by 49 W m$^{-2}$. In contrast, under moist conditions in summer the mean net surface CRE becomes more negative by 25 W m$^{-2}$, while in other seasons the mean net surface CRE was increased by 5–37 W m$^{-2}$. Trends in occurrence of dry and moist anomalies were analyzed based on 25-year-radiosonde database. Dry anomalies become less frequent with rates for different seasons from -12.8 to -4% per decade, while the occurrence of moist event increases at rates from 2.8 to 6.4% per decade. Taking into account the relations between the anomaly types and cloud properties the trends might be related to an increase in cloud occurrence, LWP, and IWP. The change in cloud properties could, in turn, modulate the surface CRE and lead to stronger surface cooling and warming related to clouds in summer and other seasons, respectively.

## 1 Introduction

It is well known that during the past 3 decades the Arctic climate has been drastically changing. The change in the annual near-surface temperature over the Arctic region has been found to be a factor of 2 to 3 larger compared to the global average (IPCC, 2007; Solomon et al., 2007; Wendisch et al., 2017). In the period 1998-2012, the temperature increase in the Arctic was persistent in contrast to the "hiatus" in the global warming having been discussed in a number of studies (Wei et al., 2016; Huang et al., 2017).

This amplification of warming in the Arctic is caused by several feedback mechanisms. Among them are the reduced sea ice extent and high sea surface temperature (Serreze et al., 2011; Hegyi and Taylor, 2018), changes in atmospheric circulation





(Maturilli and Kayser, 2017a; Overland et al., 2016; Overland and Wang, 2016; Wu, 2017) and energy transport (Graversen and Burtu, 2016; Hwang et al., 2011), surface albedo effect (Graversen et al., 2014), increased greenhouse effect of water vapor, and clouds (Yoshimori et al., 2017).

The analysis of contemporary climate models shows that on average the mean cloud feedback is one of the major mechanisms opposing the Arctic amplification from a top of atmosphere (TOA) perspective with relatively small contribution to the warming

at the surface (Pithan and Mauritsen, 2014). Modeling of the cloud impact on the Arctic amplification is still uncertain (Pithan and Mauritsen, 2014; Hwang et al., 2011; Taylor et al., 2013) due to a large number of microphysical processes (Morrison et al., 2012) and complex relations between clouds and other feedback mechanisms of the Arctic climate (Graversen and Burtu, 2016; Hwang et al., 2011). As a result some models underestimate amount of super-cooled liquid water (Sandvik et al., 2007; Nomokonova et al., 2019), which may lead to a bias in the surface temperature ranging from $-7.8$ to $0°C$ (Kay et al.,

2016; Miller et al., 2017).

Properties of Arctic clouds are significantly affected by air masses transported from the mid-latitudes (Graversen and Burtu, 2016; Hwang et al., 2011). A number of studies have already related the air transport and atmospheric rivers to amount of water vapor, cloud properties, and the radiation budget (Hwang et al., 2011; Boisvert et al., 2016; Mortin et al., 2016; Sedlar and Tjernström, 2017; Hegyi and Taylor, 2018). Hegyi and Taylor (2018) reported that the episodes of poleward atmospheric

water vapor transport are associated with periods of increased water vapor and cloud cover resulting in enhanced downwelling longwave surface fluxes and reduced surface cooling efficiency. Raddatz et al. (2013) analyzed the impact of cloud coverage and increased water vapor on longwave downwelling radiation using ground-based observations installed at different sites in the Beaufort Sea–Amundsen Gulf region of the Canadian Arctic. The authors found that the cloud coverage and water vapor explain 84% of the variance of longwave downwelling radiation, while the remaining 16% are associated with cloud

composition, cloud thickness and cloud-base height. Raddatz et al. (2013) estimated differences in longwave downwelling radiation between cases with typical and maximum values of water vapor. The authors associate the latter to moist intrusion events. The differences are 82 and 95 W m$^{-2}$ in winter and 38 W m$^{-2}$ and 45 W m$^{-2}$ in summer for clear sky and cloudy conditions, respectively. A limited number of studies show that air transportation patterns may influence phase partitioning and amount of liquid in Arctic clouds (Qiu et al., 2018; Tjernström et al., 2019).

Many studies relate the advection of moist and warm air from the mid-latitudes with Arctic climate change (Woods and Caballero, 2016; Graversen and Burtu, 2016; Johansson et al., 2017), while only a few are focused on cold intrusions (Kanno et al., 2019). However, extreme cold events exhibit a stronger change in occurrence than extreme warm events (Sillmann et al., 2013; Collins et al., 2013). Kanno et al. (2019) reported that the occurrence of extremely cold air masses in the Arctic have been reduced by about 80% over the past 60 years. The authors mention that even though the main driver of this reduction is

radiative forcing associated with green house gases, the relations of the extreme cold air masses with other components of the Arctic climate, such as humidity and cloudiness, have to be explored. Yamanouchi (2018) showed a case study with a contrast in cloud conditions and longwave radiation during a transition from cold to warm periods. The author concluded that the cold periods might be associated with low occurrence of clouds and relatively thin clouds, while high cloudiness and thick clouds





are typical for periods of warm and moist intrusions. Since the author investigated only on a short period, an analysis of longer

cloud observations is still needed.

A number of studies focus on observations at Ny-Ålesund located in the Svalbard region (Wendisch et al., 2019; Maturilli et al., 2013; Maturilli and Ebell, 2018; Yeo et al., 2018), an Arctic area where the specific synoptic regime brings more moisture from the lower latitudes in comparison to the rest of the Arctic (Dahlke and Maturilli, 2017; Mewes and Jacobi, 2019). The Svalbard region is also located in the area with the highest warming temperature trend in the Arctic (Susskind et al., 2019).

Ny-Ålesund is located at the coastline of Svalbard and thus, its climate is significantly influenced by diabatic heating from the warm ocean (Serreze et al., 2011; Mewes and Jacobi, 2019) and by the surrounding orography (Maturilli and Kayser, 2017a).

Maturilli and Kayser (2017a) have shown highly pronounced warming and moistening of the tropospheric column in Ny-Ålesund. Analyzing a 22-year radiosonde dataset (1993-2014) the authors found that in winter time there has been a significant increase of atmospheric temperature (up to 3 K per decade) and mean integrated water vapor ($+0.83\pm1.22$ kg m$^{-2}$ per decade).

This tendency in winter is correlated with a strong increase in up- and downward longwave radiation of $+11.6\pm10.9$ W m$^{-2}$ and $15.6\pm11.6$ W m$^{-2}$, respectively (Maturilli et al., 2015). In contrast, during other seasons the trends in temperature and up- and downward longwave radiation are less pronounced.

Dahlke and Maturilli (2017) reported that the observed trend in warming in the winter season is due to the increasing air mass transport through the North Atlantic pathway and reducing flow from the north. Yeo et al. (2018) investigated how the advection

of warm and cold air masses affects cloudiness, longwave fluxes at the surface and near-surface temperature at Ny-Ålesund during winter. The authors analyzed a 10-day period in February with alternating warm and cold conditions related to distinct circulation patterns. During cold periods Yeo et al. (2018) observed a reduced cloudiness and down-welling longwave flux of 200-230 W m$^{-2}$. In contrast, warm periods were associated with cloud occurrence close to 100 % and enhanced down-welling longwave flux of 300 W m$^{-2}$.

Most of the studies on the Arctic climate are focused on winter. Nevertheless, Mortin et al. (2016) and Hegyi and Taylor (2018) pointed out the importance of the representation of the atmospheric variability during the transition periods. Significant anomalies in temperature,water vapor, and cloud properties initiate the surface melt in spring, while in autumn these factors affect ice freeze-up. Maturilli et al. (2015) showed that at Ny-Ålesund the trend in the net radiation budget is highest in summer. Even though this trend does not directly translate into the surface temperature increase, the additional radiation income may

affect some other feedback mechanisms in the Arctic.

In this study, the main scientific question is how the large scale circulation and advection influence cloud appearance at Ny-Ålesund and how this influence affects the cloud radiative effect. We put our focus on periods of increased and decreased amount of integrated water vapor (IWV) and temperature, since these periods can be related to the changes in large scale circulation and synoptic situation. Even though only IWV and temperature are considered in this study, cloud formation and

development also depends on a number of other factors such as aerosol load and chemical composition (Baustian et al., 2012; Murray et al., 2012; Wex et al., 2019), dynamics (Korolev and Field, 2008; Schmidt et al., 2014), influence of surface layer (Morrison et al., 2012) and local orographic effects (Houze, 2012) and other processes. Analysis of these factors is out of the scope of this study. We used ground-based cloud observations at Ny-Ålesund. Instrumentation and data products used in this





study are presented in section 2. The definition of warm/cold and moist/dry anomalous periods is described in section 3. In
section 4 the occurrence of different types of clouds, cloud properties and their radiative effect are analyzed and related to
the anomalous periods of IWV and atmospheric temperature. Finally, the discussion of results and the summary are given in
section 5.

## 2 Instrumentation and data products

This study is based on thermodynamic and cloud measurements from a set of passive and active remote sensors continuously
running at the AWIPEV observatory, operated by the Alfred Wegener Institute Helmholtz Centre for Polar and Marine Research
(AWI) and the French Polar Institute Paul Emile Victor (PEV). Within the Transregional Collaborative Research Center (TR
172) project "Arctic Amplification: Climate Relevant Atmospheric and Surface Processes, and Feedback Mechanisms (AC)[3]"
(Wendisch et al., 2017), the instrumentation at AWIPEV observatory was complemented with a Doppler cloud radar since
June 2016. The analyzed period ranges from June 2016 to October 2018, when continuous cloud radar observations were
available at AWIPEV observatory. In addition to ground-base observations, we also used a number of observation and modeling
products characterizing clouds, air transportation, and radiation budget. Most of instrumentation and products are the same as
in Nomokonova et al. (2019) and, therefore, only briefly described in this study.

### 2.1 Microwave radiometer observations

The humidity and temperature profiler (HATPRO; Rose et al., 2005) has been operating at the AWIPEV station since 2011.
HATPRO is a fourteen-channel microwave radiometer that measures brightness temperatures (TB) at K-band (22.24-31.40 GHz)
and at V-band (51.26-58 GHz) frequencies with a temporal resolution of 1-2 s. The TBs measured at K-band are used for re-
trievals of the integrated water vapor (IWV), liquid water path (LWP) and humidity profiling. The V-band channels are located
along the oxygen absorption complex at 60 GHz and are used for vertical temperature profiling. For this study we used tem-
perature profiles, IWV, and LWP retrieved as described in Löhnert and Crewell (2003). The retrievals were recently adapted
for an operation at Ny-Ålesund (Nomokonova et al., 2019). HATPRO measures continuously but cannot provide reliable in-
formation during rain conditions when the radome of the instrument is wet. In these cases, data are flagged and excluded from
the analysis.

### 2.2 Ceilometer

Since 2011 a Vaisala ceilometer CL51 has been operated at the AWIPEV observatory (Maturilli and Ebell, 2018). The ceilome-
ter emits pulses at 905 nm wavelength and measures atmospheric backscatter with a temporal resolution of about 10 s and a
vertical resolution of 10 m. The maximum profiling range is 15 km.

The ceilometer is sensitive to the surface area of the scatterers and is thus strongly affected by high concentrations of particles
like cloud droplets and aerosols (Hogan et al., 2006). On the one hand, it is thus well suited to detect liquid layers and cloud
base heights. On the other hand, the near-infrared signal is significantly attenuated by liquid layers. Therefore, the ceilometer





often cannot detect cloud particles above the lowest liquid layer when optical depth exceed a value of around 3. Therefore, for this study we used a cloud base height which is the lowest altitude of cloud boundary detected by ceilometer.

## 2.3 Cloudnet products

The Cloudnet algorithm suite (Illingworth et al., 2007) running at Ny-Ålesund combines observations from a Doppler cloud radar, Vaisala ceilometer CL51 (Maturilli and Ebell, 2018), HATPRO, and thermodynamic profiles from a NWP model.

In this study we used the target categorization, which is a standard product of Cloudnet. This product has a temporal and vertical resolution of 30 s and 20 m, respectively. The Cloudnet categorization was used for the cloud classification as described in Nomokonova et al. (2019). In addition, based on the target categorization for cloud regions with ice particles we calculated the ice water content (IWC) based on radar reflectivity factor Z and environment temperature according to Hogan et al. (2006). More detailed description of the used Cloudnet products and their uncertainties is given in Nomokonova et al. (2019).

## 2.4 Surface cloud radiative effect

The surface cloud radiative effect (denoted as CRE throughout the study), which will be analyzed for the different humidity conditions, was derived from the broadband radiative transfer calculations by Ebell et al. (2019) with the rapid radiative transfer model RRTMG for the analyzed period at Ny-Ålesund. The model provides vertically resolved shortwave (SW) and longwave (LW) up- and downward fluxes and heating rates. In this study CRE is calculated as follows:

$$\mathrm{CRE} = (F_\downarrow - F_\uparrow) - (F_{\downarrow clr} - F_{\uparrow clr}), \tag{1}$$

where $F_\downarrow$ and $F_\uparrow$ are down- and upwelling all-sky fluxes at the surface, respectively. $F_{\downarrow clr}$ and $F_{\uparrow clr}$ are surface down- and upwelling fluxes, which would be if the sky were cloud free. LW, SW, and net fluxes from the model are used in Eq. 1 for the calculation of LW, SW, and net CRE, respectively. Ebell et al. (2019) estimated the uncertainties in CRE using 10-min averaged fluxes observed by the baseline surface radiation network (BSRN). The uncertainties in CRE depend on averaging time. For

time periods ranging from days to month, which are analyzed in this study, the uncertainties are estimated to be smaller than 6.4, 2.0, and 6.7 W m$^{-2}$ for SW, LW, and net CRE, respectively.

## 2.5 Backtrajectories FLEXTRA

In order to analyze which patterns in air transportation are related to episodes of warm/cold and moist/dry conditions at Ny-Ålesund, we used the output of the 3-dimensional FLEXTRA trajectory model version 3.0 (Stohl et al., 1995; Stohl and Seibert,

1998; Stohl, 1998). The calculations of the trajectories are based on data of the European Centre for Medium range Weather Forecast (EMCWF) with the initialized analyses every 6 hours and horizontal resolution of 1.125°. The temporal resolution of the back trajectories is 3 hours. We used FLEXTRA files (see https://projects.nilu.no//ccc/trajectories/evdc/ for detailed information) generated for the Zeppelin station (78.9°N 11.88°E), the arrival height of 1500 m, and going 6 days back. Since Ny-Ålesund is surrounded by up to 1000-m high mountains, the arrival height of 1500 m altitude was chosen in order to avoid

orographic effects in the large scale air transport.





## 2.6 Radiosonde observations

Radiosondes at the AWIPEV station have been launched at least once per day since 1993 (Maturilli and Kayser, 2017a). The radiosonde data since June 2006 have been routinely processed by GRUAN version 2 data processing algorithm (Sommer et al., 2012; Maturilli and Kayser, 2017a). More details on the radiosonde dataset can be found in Maturilli and Kayser (2016, 160  2017b). In the present study the radiosonde data for the period from 1993 to 2018 were used to analyze the long-term changes in occurrence of moist and dry conditions, respectively.

## 3 Identification of periods with increased/decreased moisture and temperature

In this study we analyze 6-hour periods with distinct values of temperature and IWV. Throughout the study periods with decreased and increased values are denoted by "-" and "+" signs prior to a variable symbol, respectively (e.g. "+T" corresponds 165  to periods with increased temperature), while typical values are indicated just by a variable symbol (e.g. "T").

In order to decide whether a period is associated with a particularly high (low) value of water vapor and temperature, we use a dataset of 6-hourly mean values of IWV and temperature at 1450 m from HATPRO for the period from 2011 to 2018 (this period is used as reference throughout the study). The 1450 m HATPRO range bin was chosen to account for the large scale transport rather than local effects which are related to the orography around Ny-Ålesund (Maturilli and Kayser, 2017a). 170  This altitude is also the closest to the arrival height of the FLEXTRA back trajectories. Since the atmospheric conditions vary throughout a year, we calculated the 10th and 90th percentiles of the temperature and IWV for each month using the reference dataset (Fig. 1). The percentiles of IWV and 1450 m temperature are used as thresholds for the event classification. If a 6h-period has a mean IWV value below the 10th or above the 90th percentile it is considered as dry ("-IWV") or moist ("+IWV"), respectively. Otherwise it is assumed that the value of water vapor of this period is normal ("IWV"). Similarly, "-T" and "+T" 175  correspond to a period with an average 1450 m temperature below the 10th or above the 90th percentile, respectively. As many studies consider water vapor as a driver of changes in radiative properties of the Arctic atmosphere, we use IWV as an indicator of anomalous periods. Typical periods corresponding to the "IWV" class are further denoted as normal. An anomalous period is one with "-IWV" (dry anomaly) or "+IWV" (moist anomaly), regardless which temperature class the period has. In addition we also analyze periods when the water vapor anomalies are supported by temperature anomalies, i.e. classes "-T -IWV" and 180  "+T +IWV".

Since the anomalies in temperature and IWV are often driven by certain weather patterns which are related to the transport of air masses coming from lower or higher latitudes (Maturilli and Kayser, 2017a; Dahlke and Maturilli, 2017; Mewes and Jacobi, 2019; Wu, 2017), we analyzed back trajectories for all dry and moist anomalies. We identified all 6-hourly periods with an anomaly class within the analyzed period and found FLEXTRA back trajectory files with corresponding reaching time. 185  Figure 2 shows 6-day-trajectories with the end point in Ny-Ålesund for "-IWV" and "+IWV" anomalies in different seasons. 6-day trajectories are sufficient to capture the air transportation to the Arctic. Graversen (2006) found a correlation between intensity of the atmospheric northward energy transport across 60°N and the Arctic warming/cooling with the time lag of about 5 days. As expected, the occurrence of moist anomalies ("+IWV", red lines in Fig. 2) is associated with the air transport from





the south, while the dry anomalies ("-IWV", blue lines in Fig. 2) related to the air coming from the north. There is a slight
difference between the seasons. In winter and spring, dry air typically circulates counterclockwise, from North of Russia over
the North Pole region and northern Greenland. Mewes and Jacobi (2019) have shown that this kind of circulation happens
when air from the North Pacific flows into the Arctic. In summer, dry anomalies are mostly associated with air coming from
northern Canada and Greenland. In autumn, there are two distinct pathways, from south-east and west, although the "-T -IWV"
anomalies are related to air coming predominantly from the west (not shown). Wet anomalies are mostly driven by the air
advected from the North Atlantic. In autumn and summer, a significant part of moist events originates in the Scandinavian
region and Barents sea. The transport pathways for "-T -IWV" and "+T +IWV" events (not shown) are in general similar to
those of "-IWV" and "+IWV", respectively, which is in agreement with the results found by Mewes and Jacobi (2019), who
showed that the air transport from the North Atlantic sector is typically associated with the positive temperature anomaly, while
the transport from Siberia and the North Pacific is connected to a negative temperature anomaly in the Arctic.

Table 1 summarizes the occurrence of different types of periods for the analyzed period from June 2016 to October 2018.
The occurrence of moist anomalies in winter and summer for the analyzed period is nearly the same as for the reference period,
when, according to our definition, the occurrence of moist and dry anomalies was 10%. In spring and autumn the occurrence
is 8 and 14.2%, respectively. The increase in occurrence of moist anomalies during the polar-night season of 2016-2017 was
recently reported by (Hegyi and Taylor, 2018). The authors analyzed the whole Arctic region and related the increase to more
frequent moisture intrusions from the Atlantic and Pacific regions. Our results show that the occurrence of dry anomalies is
about 8% for winter and autumn. In spring and summer dry anomalies were observed about 13% of time. Periods with "+T
+IWV" anomaly take a major part (about 67%) of all moist anomalous cases. In contrast, occurrence of "-T -IWV" periods is
only about 35% of all dry anomalies in all seasons except winter, when the occurrence is almost 90%. Thus, the dry anomalies
are not regularly accompanied by a negative anomaly in temperature, while the opposite is valid for moist anomalies.

## 4   Results and discussion

In this section we show how the anomalous conditions are related to cloud macro- and microphysical parameters such as
occurrence, type, phase partitioning, and liquid and ice water path (IWP) and cloud radiative properties. For the characteri-
zation of clouds we apply the cloud detection and classification method based on Cloudnet that was previously described in
Nomokonova et al. (2019). Note, that the method is applied on Cloudnet profiles with no liquid precipitation.

### 4.1   Cloud occurrence

Figure 3 shows the frequency of occurrence (FOC) of different cloud types during anomalous and normal conditions. Clouds
are present in 70–80% of cases with normal values of IWV. Among them about a half are multi-layer clouds.

In dry anomalous events, the FOC of clouds is in general lower and ranges from 26% in spring to 70% in summer. The
decrease in FOC of clouds is mostly caused by less frequent multi-layer clouds (MC), whose occurrence in "-IWV" conditions
drops by a factor of 2 to 4. During spring and autumn clouds are about a factor of two more frequent during "-T -IWV" events





than in "-IWV" cases. The enhanced FOC of clouds may be due to a higher probability of cloud particle formation at lower temperatures for a given amount of water vapor. Nevertheless, during winter and summer there is less difference in cloud occurrence between "-IWV" and "-T -IWV" events.

Unexpectedly high occurrence of clouds (∼92%) was found during "-T -IWV" episodes in autumn. We found that all "-T -
IWV" events in autumn occurred from 26 to 30 September 2018. Such a short time period would probably not be representative for autumn "-T -IWV" cases if a longer dataset were analyzed. According to the FLEXTRA back trajectories for this time period, air was primarily transported from the northern Greenland area. As it will be shown below, the "-T -IWV" episodes in autumn were also characterized by LWP and IWP values exceeding those under normal conditions. A deeper understanding of this phenomenon requires further investigations, which are out of the scope of this study.

Higher availability of water vapor ("+IWV") leads to an increase in FOC of clouds up to 90–99%. The increase is mostly caused by changes in MC, while the FOC of single-layer clouds (SC) is not much affected. As it was mentioned in Sec. 3, moist anomalies are often accompanied by positive temperature anomalies. Therefore, differences in cloud occurrence between "+IWV" and "+T +IWV" events are small. Our findings are in agreement with the study by Gallagher et al. (2018) for Greenland. They showed that during atmospheric circulations associated with increased (decreased) moisture, the number of clear
sky scenes reduces (increases).

## 4.2   Cloud phase

Cloud ice and liquid have distinct microphysical properties. For instance, the size of ice particles is in general larger than for liquid droplets while the latter have a higher number concentration (Korolev et al., 2003). Ice particles can have a large variety of shapes (Bailey and Hallett, 2009). In addition, liquid water and ice have different dielectric properties (Ray, 1972). Thus, the
phase composition of clouds affects SW and LW radiative properties of clouds (Ebell et al., 2019). Therefore, we also analyzed the occurrence of different types of hydrometeors in the atmospheric column (Fig. 4).

In general, profiles with liquid phase (sum of green and orange columns in Fig. 4) occur more often during moist periods and less often during dry periods. The FOC of liquid containing profiles during "+IWV" and "-IWV" was characterized by the change of more than +30 and -30% relative to normal conditions. Only in summer the increase in FOC of liquid containing
profiles between moist and normal periods is 8%. The increase in FOC of clouds in "+IWV" events was mostly related to higher occurrence of MC. FOC of liquid containing clouds during "+IWV +T" and "-IWV -T" anomalies do not differ much from the corresponding water vapor anomalies in all seasons except autumn, when all events corresponded to the single continuous 5-day episode with air masses transported from northern Greenland mentioned in Sec. 4.1.

Ice containing profiles (sum of blue and green columns in Fig. 4) occur more often under "+IWV" and "+IWV +T" conditions
and less during "-IWV" and "-IWV -T". The change in ice containing profiles is mostly defined by the change in profiles with both liquid and ice (green columns in Fig. 4), since FOC of pure ice phase (blue columns in Fig. 4) varies only slightly with change in IWV. Gallagher et al. (2018), who investigated the influence of atmospheric circulations on cloud composition in Greenland, similarly showed that moist/dry conditions lead to increase/decrease in occurrence of mixed-phase clouds, which





are the dominant type of liquid containing clouds in the Arctic (Shupe et al., 2006). Gallagher et al. (2018) also noted that ice
clouds are not constrained by circulation types to the same degree as mixed-phase clouds.

## 4.3 Liquid and ice water path

Moist and dry anomalies influence not only the cloud occurrence and composition but also water content. We therefore link
LWP and IWP to the different types of anomalies (Fig. 5). Note, that LWP and IWP were only calculated for liquid-containing
and ice-containing profiles, respectively. Clear-sky cases were not added for calculations of the statistics of LWP and IWP.
Following the changes in occurrence of liquid- and ice-containing clouds, LWP and IWP increase under moist conditions
and decrease under dry conditions. "+IWV" anomalies cause an increase in mean LWP by a factor of 1.5–2.0 relative to normal
conditions and also increase the variability in LWP. During dry anomalies, LWP is significantly lowered and does not exceed
12 and 94 g m$^{-2}$ in winter/spring and summer/autumn periods, respectively. Gallagher et al. (2018) recently showed that
atmospheric circulation types associated with enhanced water vapor in Greenland often lead to increased LWP. The authors
also showed that opposite is valid for dry conditions, when decreased values of LWP are more likely. Note, that the unexpected
two-fold LWP increase during the "-T -IWV" periods in autumn correspond to the 5-day episode with air mass transported
from northern Greenland mentioned in Sec. 4.1. Moist anomalies correspond to an increase in mean IWP by a factor 3 relative
to normal conditions in winter and spring, and by a factor of 2 in summer and autumn. In winter and spring, dry conditions
decrease mean IWP values by an order of magnitude, which may be related to a strong reduction in occurrence of ice containing
clouds and less efficient ice particle growth during "-IWV" events. In contrast, during summer and autumn, mean IWP is
reduced by a factor of 1.3.

Thus, the results reveal a strong impact of the anomalous periods on LWP and IWP and, in particular in winter and spring.
Even though mean IWP values during normal conditions are nearly the same in winter and autumn, relative changes in IWP
caused by dry and moist anomalies differ drastically among the two seasons. In winter, wet and dry conditions lead to a 3-fold
increase and 10-fold decrease, respectively. In contrast, the increase in autumn is a factor of 2 and there is almost no decrease.
Since the anomaly type cannot fully explain this effect, it is probably also related to other factors. One of such factors could be
aerosols. Weinbruch et al. (2012); Lange et al. (2018); Jung et al. (2018); Wex et al. (2019) have shown that concentration and
chemical composition of cloud condensation (CCN) and ice nuclei (IN) also have a seasonal variability in the Arctic region.
Dall´Osto et al. (2018) and Lange et al. (2018) found that accumulated aerosol mode is dominant in winter season, while in
summer ultrafine aerosol population becomes more abundant. Jung et al. (2018) showed that the seasonal change in the aerosol
type affects the activation ability of CCN in the Svalbard region and found the highest activation efficiency in winter and lowest
in summer, with intermediate values in spring and autumn. Wex et al. (2019) investigated the annual cycle of IN particles in
different Arctic regions. The authors found that IN concentration is often an order of magnitude higher in summer and autumn
than in winter and spring. Note, that within this study aerosols are not analyzed.





## 4.4 Surface cloud radiative effect

In the previous sections, we showed the changes of cloudiness and amount of liquid and ice in a column under various atmospheric states. Liquid-containing and pure ice clouds have a different impact on the radiation budget at Ny-Ålesund (Ebell et al., 2019) and occurrence of these types of clouds varies for dry and moist conditions. Therefore, we also estimated the surface CRE under different atmospheric conditions. Figure 6 summarizes the surface SW, LW, and net CRE for different anomaly periods.

Values of LW CRE are in the range from 0 to 85 and correlate with the cloud occurrence and amount of liquid in a column (Ebell et al., 2019). The large variability of LW CRE distributions (Fig. 6a) is explained by their bimodality (Shupe and Intrieri, 2004): clear sky cases and profiles with ice only are characterized by no or low LW CRE, while for liquid containing clouds LW CRE is typically high with values from 50 to 85 W m$^{-2}$ (Ebell et al., 2019). The upper limit of LW CRE corresponds to clouds with LWP > 50 g m$^{-2}$ and/or IWP > 150 g m$^{-2}$ (Miller et al., 2015; Ebell et al., 2019).

During moist anomalies the mean LW CRE increased to 60–70 W m$^{-2}$ in winter, spring, and autumn. Thus, the mean LW CRE, enhanced under moist anomalies in winter, spring, and autumn, can exceed the typical mean LW CRE in summer (Fig. 6b, white box). This increase in mean LW CRE is associated with high cloud occurrence, which mostly exceeds 90% under moist conditions. In addition "+IWV" cases typically characterized by high mean LWP and IWP exceeding 90 and 200 g m$^{-2}$, respectively. In summer the mean LW CRE during moist anomalies becomes lower than under normal conditions. This effect may be caused by influence of water vapor in presence of optically thick clouds as was previously described by Cox et al. (2015). The authors found that for higher relative humidity LW CRE is typically lower. Ebell et al. (2019) identified a similar effect at Ny-Ålesund, where the decrease in LW CRE at the surface for higher IWV corresponds to clouds with LWP exceeding 300 g m$^{-2}$. In general, relative humidity at Ny-Ålesund is high in summer (Maturilli and Kayser, 2017a; Nomokonova et al., 2019). Moreover, in "+IWV" cases we would expect even higher values of relative humidity, since on average there are no positive temperature anomalies relative to normal conditions.

Dry anomalies correspond to a reduction of the mean LW CRE to 5–11 W m$^{-2}$ in winter and spring and to 29–32 W m$^{-2}$ in summer and autumn. Hence, the dependence of the mean LW CRE on IWV is more pronounced in winter and spring than in autumn and especially in summer. Such behavior though is not directly related to changes in IWV itself but rather to coupling of the changes in IWV to cloud properties. As it was previously shown, dry anomalies are associated with reduced cloud occurrence, amount of liquid-containing clouds, LWP, and IWP, while increased values of these parameters are related to moist periods. In summer the cloud properties do not show as strong change during dry anomalies with respect to normal conditions as in other seasons, which may reflect into the smaller corresponding change in LW CRE.

Besides the influence of cloud occurrence and microphysical properties, the LW CRE also depends on altitude at which clouds occur. Shupe and Intrieri (2004); Dong et al. (2010) showed that LW CRE increases with decreasing cloud base height (CBH). Shupe and Intrieri (2004) show that Arctic clouds with bases below and above 3 km have median LW CRE around 45 and 20 W m$^{-2}$, respectively. Figure 7 shows CBH measured by the ceilometer for the analyzed period. Note, that due to the instrument limitations (see section 2.2), which are related to the attenuation of the ceilometer signal in optically thick clouds,





only the lowest CBH was taken into account. Throughout a year, CBH is mostly below 2 km. During moist and dry anomalies

CBH either does not change or slightly decreases, which may cause an increase in LW CRE. Although, Yeo et al. (2018), who

analyzed the dependence of LW fluxes measured at Ny-Ålesund on CBH, showed that the mean LW CRE of clouds within

the lowest 2 km does not differ by more than 10 W m$^{-2}$. Only dry anomalies in spring and moist anomalies in summer are

related to higher CBH. Taking into account low cloud occurrence during dry conditions in spring the increase in CBH should

not change LW CRE much. In summer the increase in CBH during moist conditions could be another factor (in addition to the

effect of water vapor described above) preventing an increase in LW CRE.

The SW CRE is only significant when the sun is above the horizon. Thus, the strongest SW CRE can be found in summer.

Under normal conditions in summer and spring the mean SW CRE is -115 and -19 W m$^{-2}$. An absolute change in CRE$_{\text{SW}}$

can in general be caused by three main factors: cloud properties, solar zenith angle (SZA), and surface albedo ($\alpha$). The 6-fold

difference in the mean SW CRE in spring and summer might be associated with the changes only in surface albedo and SZA,

since FOC of clouds, LWP, and IWP vary only slightly.

A number of studies have already shown that the surface albedo under clear sky and cloudy conditions can alter the SW

CRE at the surface (Shupe and Intrieri, 2004; Miller et al., 2015; Miller et al., 2017; Ebell et al., 2019). Ebell et al. (2019)

found that at Ny-Ålesund the surface albedo (ratio between upward and downward SW fluxes at the surface retrieved by the

RRTMG) exceeds 0.8 when the surface is covered by snow and is below 0.15 in bare tundra. For the analyzed period, the

change in the mean surface albedo between normal conditions and anomalies does not exceed $\pm 0.05$ in spring and $\pm 0.1$ in

summer and autumn. Shupe and Intrieri (2004) showed that SW CRE is nearly proportional to $1 - \alpha$. The changes in SW CRE

between normal and anomalous conditions caused by the variability in the surface albedo do not exceed 30% in spring and

15% in summer and autumn.

In addition to the surface albedo, the SW CRE at the surface depends on SZA (Minnett, 1999; Miller et al., 2018). Ebell

et al. (2019) showed that in Ny-Ålesund the lowest SZA, which corresponds to the highest position of the sun, is in summer

with the minimum of 55° in June. In spring values of SZA are larger. Shupe and Intrieri (2004) have found that for SZA higher

than 70° the shortwave cooling effect of clouds decreases due to a rapid decrease of solar insolation. When the sun is high in

the sky, the shading effect of the clouds becomes stronger (Shupe and Intrieri, 2004).

There are two ways SZA can influence differences in SW CRE between anomalous and normal cases: (1) an anomaly can

have a dominant daytime of occurrence, while normal conditions are uniformly distributed over a day, and (2) an anomaly

can concentrate in a certain part of a season. In order to check for changes in SW CRE between anomalous and normal cases

caused by diurnal cycles of SZA we checked whether anomalies have a dominant time of occurrence. We found that for the

analyzed period normal and anomalous cases were nearly uniformly distributed among 6-hourly periods and therefore time of

a day is neglected in the following analysis. In order to mitigate the remaining effects of SZA due to the sparse distribution of

anomalies over seasons we adopt the approach of Sengupta et al. (2003) and calculate the normalized SW CRE at the surface:

$$\text{nCRE}_{\text{SW}} = \frac{\text{CRE}_{\text{SW}}}{F_{\downarrow \text{clr,SW}} - F_{\uparrow \text{clr,SW}}}, \tag{2}$$





where $CRE_{SW}$ is the surface SW CRE, and $F_{\downarrow clr,SW}$ and $F_{\uparrow clr,SW}$ are down- and upwelling SW fluxes at the surface that would be if the sky were cloud free.

Under normal conditions the mean $nCRE_{SW}$ is -0.2 and -0.45 in spring and summer, respectively. Dry conditions increase
the mean $nCRE_{SW}$ to nearly 0 and -0.2 in spring and summer, respectively. In moist events the mean $nCRE_{SW}$ decreases below –0.4. Since the effects of SZA are mitigated in $nCRE_{SW}$, the variability in mean $nCRE_{SW}$ for different anomalies is mostly defined by cloud properties and the surface albedo.

The relative change in $CRE_{SW}$ is unsusceptible to differences in SZA but only when anomaly cases are uniformly distributed over a season. For anomaly cases uniformly distributed over a season we also expect no difference in surface albedo between
anomaly and normal conditions. Thus, similar relative changes in $CRE_{SW}$ and $nCRE_{SW}$ and near-zero change in the surface albedo indicate similar SZA and surface albedo for anomaly and normal cases, while the diversion would show that anomaly cases were sparsely distributed over a season.

In Table 2 we summarize changes in $CRE_{SW}$, $nCRE_{SW}$, and the surface albedo related to dry and moist anomalies. The results show that dry cases are related to positive absolute changes in SW CRE, thus pointing to less efficient SW surface
cooling by clouds relative to normal conditions. The largest difference of 67.2 W m$^{-2}$ is in summer, when the cloud can produce the strongest SW shading (Shupe and Intrieri, 2004; Ebell et al., 2019). In spring and summer $CRE_{SW}$, $nCRE_{SW}$ change by nearly the same factor and the absolute change in the surface albedo does not exceed 0.05. Similar relative changes in $CRE_{SW}$, $nCRE_{SW}$ and near-zero absolute change in the surface albedo indicate that the difference in the mean SW CRE is mainly caused by changes in cloud properties and not by SZA. In autumn $CRE_{SW}$ changed by –94% while $nCRE_{SW}$ changed by
–60%. The difference in the relative changes indicates that it is caused by SZA because dry autumn cases were not uniformly distributed over the season. The absolute change in the surface albedo was 0.45. Thus, the difference in SW CRE between dry and normal cases under dry conditions in autumn were caused by all three factors, i.e. cloud properties, SZA, and the surface albedo. The direction of the changes in $CRE_{SW}$ is consistent with Shupe and Intrieri (2004), who showed that an increase in the surface albedo corresponds to a reduction in the cloud induced surface SW cooling.

During moist anomalies SW CRE is more negative than for normal conditions. In summer the relative changes in $CRE_{SW}$ and $nCRE_{SW}$ are close and the surface albedo is not altered by more than 0.06. Therefore, we conclude that the change in the mean SW CRE of –25.6 W m$^{-2}$ is mainly caused by cloud properties. In spring and autumn the absolute change in the surface albedo is also relatively low. Nevertheless, the relative changes in $CRE_{SW}$ and $nCRE_{SW}$ differ. This indicates that the absolute change in the mean SW CRE is likely caused not only by cloud properties but also by SZA.

Gallagher et al. (2018) showed an effect of moist and dry anomalies on CRE. The authors analyzed the Summit site in Greenland, where typical IWV values are in the order of 1–3 kg m$^{-2}$ (Pettersen et al., 2018), which is drier than at the Ny-Ålesund station (see Fig. 1). Gallagher et al. (2018) used data from 2011 to 2015. The authors found that the southern transport pattern, associated with increase in IWV by 0.69 kg m$^{-2}$, leads to a change in LW and SW CRE of +13 and –3 W m$^{-2}$, respectively, relative to corresponding typical values. Since Gallagher et al. (2018) did not analyze seasons separately, in
order to compare the results, we also averaged our results over the whole analyzed period. For Ny-Ålesund we found mean differences between "+IWV" and "IWV" for LW and SW CRE of +21.2 and -10.2 W m$^{-2}$, respectively. In contrast, northern





circulation pattern in Greenland leads to decrease in IWV by 0.34 kg m$^{-2}$ changes LW and SW CRE by -6.1 and 0.1 W m$^{-2}$ (Gallagher et al., 2018). Our results for Ny-Ålesund are -21.5 and 21.6 W m$^{-2}$, respectively. Note, that the absolute values may differ because of more humid environment in Svalbard. Nevertheless, the sign of the LW CRE change is the same. SW
CRE values are difficult to compare due to and possible differences in SZA and surface albedo. For instance, the change in SW CRE at the Summit station could be closer to 0 due to high albedo in Greenland throughout a year, while at Ny-Ålesund the surface albedo is less than 0.15 in summer (Ebell et al., 2019).

Figure 6c depicts a relation of the net CRE to anomaly types. As was reported by Curry et al. (1996), due to the absence of sunlight in the Arctic region during the polar night the LW CRE is dominant and the Arctic clouds warm the surface.
Therefore, the mean net CRE in autumn and winter is mostly defined by LW CRE. Influences of water vapor anomalies have been previously discussed in details.

In summer and spring both LW and SW contribute to the mean net CRE resulting in -71.5 and 24.7 W m$^{-2}$ under normal conditions, respectively. Dry conditions correspond to less positive LW and less negative SW CRE, which lead to the mean net CRE of -19 and 4.3 W m$^{-2}$ in summer and spring, respectively. During moist periods LW CRE increases relative to normal
conditions in spring and does not change much in summer, while SW CRE becomes more negative in both seasons. Thus, the mean net CRE under moist conditions in spring changes to 31.5 W m$^{-2}$ and to –101 W m$^{-2}$ in summer.

### 4.5 Trends in anomaly occurrence

Since IWV anomalies show a strong impact on cloud properties and their radiative effect, the next question is how the occurrence of dry and moist conditions at Ny-Ålesund has changed in the last decades. The MWR observations at Ny-Ålesund are
only available since 2011, and, therefore, cannot be used for such a long-term analysis. Instead, the radiosonde observations were used for the estimation of the occurrence of "-IWV" and "+IWV" events for the time frame from 1993 to 2018. Note, that 6-hour averaged values of IWV from MWR used for the previous analysis cannot be obtained from radiosondes. Therefore, we estimate the IWV value for each radiosonde profile and classify the profile using the thresholds defined in Sec. 1. Even though, results obtained from a radiometer would probably have been slightly different, the radiosondes still show a tendency in the
IWV anomalies.

Figure 8 shows changes in anomaly occurrences. According to a two-sided t-test, dry and moist anomalies in all seasons show significant trends with the 95% confidence level except for moist anomalies in spring. Dry anomalies have significant negative trends in all seasons, which are especially pronounced in autumn and winter with values of -10.7 and -12.9% decade$^{-1}$, respectively. About a half of the profiles in autumn and winter of 1993 corresponded to dry anomalies. These trends might be
associated with changes in atmospheric circulation found by Dahlke and Maturilli (2017) for the Svalbard region. The dry events in spring and summer also exhibit negative trends but the decrease of their occurrence is at lower rates of –6.8 and –4% decade$^{-1}$, respectively. The highest rate of trends for "+IWV" cases were found for winter and autumn with slopes of 5.6 and 6.4% decade$^{-1}$, respectively. Our results are in line with Mewes and Jacobi (2019), who showed that in winter the occurrence of North Atlantic and North Pacific pathways has increased and decreased, respectively. The North Atlantic and





North Pacific air transports are associated with increased and decreased surface temperature and IWV in the Svalbard region (Dahlke and Maturilli, 2017).

Taking into account the link between anomaly types, cloud properties, and CRE (Figs. 3–6), we conclude that during the last 25 years the changes in the occurrence of dry and moist anomalies at Ny-Ålesund may have lead to an increase in cloud occurrence, LWP, and IWP in all seasons. In turn, this could have enhanced the cloud related surface warming in autumn,
winter, and spring but produce stronger cooling in summer. If the trends of anomaly occurrence continue in the future, we expect that CRE will become more positive in autumn, winter, and spring and more negative in summer.

## 5   Summary and conclusion

This study is devoted to the analysis of anomalous, in terms of IWV and temperature at 1450 m altitude, atmospheric conditions at Ny-Ålesund. The main focus is on the impact of anomalous conditions on cloud properties and their CRE. Within this work,
anomalies are defined as a deviation of 6-hour averaged IWV and/or temperature below and above 10th- and 90th percentile of the corresponding parameter over the reference period from 2011 to 2018. Different anomaly types were related to air flows using back-trajectories FLEXTRA and cloud observations from Cloudnet. The output of the rapid radiative transfer model recently applied by Ebell et al. (2019) to the Ny-Ålesund observations was used to associate the anomaly types to a variation in CRE.

A number of studies on anomalous conditions in the Arctic concentrate only on moist intrusions and/or cover only the winter season (Woods and Caballero, 2016; Graversen and Burtu, 2016; Johansson et al., 2017; Hegyi and Taylor, 2018). In this study we focus not only on warm and moist events, but also on dry and cold anomalies, which have recently been shown to have an impact on the Arctic climate (Sillmann et al., 2013; Collins et al., 2013; Kanno et al., 2019). The study also covers all seasons since, for example, surface melt and ice freeze-up in transition periods strongly depend on anomalies in temperature and water
vapor (Mortin et al., 2016; Hegyi and Taylor, 2018). The main findings of this work are listed below:

(1) The periods of positive and negative anomalies in temperature and water vapor are correlated with large-scale air transport. Most of the moist events are associated with air flow from the North Atlantic, while dry periods are mainly caused by air circulating in the Arctic region. This finding is in agreement with previous studies (Maturilli and Kayser, 2017a; Dahlke and Maturilli, 2017; Wu, 2017; Mewes and Jacobi, 2019). An analysis of seasonal variability of transport pathways shows that in
autumn and summer a significant part of moist events originates from the Scandinavian region and Barents sea. In winter and spring dry conditions were associated with counterclockwise air circulations over the North Pole region. This is in agreement with the results from Mewes and Jacobi (2019), who showed that in winter this type of circulation is related to the North Pacific pathway, which causes cold anomalies for the Svalbard region. In autumn, two distinct pathways, leading to dry conditions at Ny-Ålesund, have been found: from South-East and West. The latter brought the 5-day "-T -IWV" episode observed from 26
to 30 September 2018 with unexpectedly high (typically the FOC of cloud is low during dry anomalies) occurrence of clouds.

(2) 67% of moist anomalies are accompanied by strong temperature increase, while only 43% of dry cases correspond to cold anomalies.





(3) Anomalies in IWV correlate with FOC of clouds. In general, dry anomalies are related to cloud occurrence ranging from 26% in spring to 70% in summer, which is on average more than 30% lower than during normal conditions. We found, that dry conditions also show FOC of multi-layer clouds decreased by a factor of 2 to 4. Although, for autumn and spring FOC of clouds was 2 times higher for "-T -IWV" events than for "-IWV" cases, which is probably due to higher likelihood of cloud particle formation at lower temperatures for a given amount of water vapor. During the moist periods, the FOC of clouds increases up to 90–99%. This increase is mainly caused by more frequent multi-layer clouds while FOC of single-layer clouds is almost not affected. In contrast to dry anomalies, the occurrence of clouds between "+IWV" and "+T +IWV" events does not show a large difference because most of the time moist anomalies were accompanied by the periods with the positive temperature anomaly.

(4) "-IWV" events are characterized by 30% relative decrease in FOC of profiles containing both ice and liquid with respect to normal conditions. This type of profiles becomes 30% more frequent under moist conditions relative to normal conditions. Profiles with only ice or only liquid are affected by water vapor anomalies to a lesser degree.

(5) Excess and shortage in water vapor has been found to be correlated with mean LWP and IWP. During winter and spring, "+IWV" events are related to a factor of 2–3 increase in LWP and IWP relative to normal conditions, while dry anomalies lead to a reduction of LWP and IWP by an order of magnitude. For example, during normal conditions the mean IWP in autumn is nearly the same as in winter. Nevertheless, the relative change in mean IWP during dry and moist events in spring does not exceed a factor of 2. Thus, the difference between winter/spring and summer/autumn cannot be explained only by water vapor anomalies and should be related to other feedback processes (e.g. difference in aerosol load, orographic effect, dynamics and etc.). During dry anomalies in summer mean LWP decrease by a factor of 2, while mean IWP does not change much. In autumn mean LWP and IWP increase by 30% during dry anomalies. Under moist anomalies mean LWP and IWP increase by a factor of 1.5 and 2.3, respectively. In autumn LWP and IWP increase by a factor of 2 during moist conditions.

(6) Dry(moist) anomalies are associated with less(more) cloud related surface SW cooling. In spring and summer during dry anomaly periods the mean SW CRE was higher. Relative to the normal conditions the changes were 19 and 67 W m$^{-2}$ in spring and summer, respectively. The higher values are associated with lower cloudiness and LWP in dry cases. The difference between summer and spring is caused by the variability of SZA and the surface albedo. During moist periods in spring and summer the cloud related cooling is enhanced by 25 W m$^{-2}$ compared to normal conditions.

(7) The mean LW CRE at the surface is reduced during dry anomalies with respect to normal cases by 25–35 W m$^{-2}$ in winter and spring, and by 11–19 W m$^{-2}$ in summer and autumn. In contrast, moist periods are related to an increase of the mean LW CRE in comparison to normal conditions. The increase was observed in all seasons except summer. For instance, in winter, spring and autumn the mean LW CRE raises from 35–41 W m$^{-2}$ under normal conditions to 64-71 W m$^{-2}$ during moist events. Thus, in winter the mean LW CRE during moist periods was even higher than the typical value in summer (51 W m$^{-2}$). In summer mean LW CRE did not change during "+IWV" periods and decreased by 6 W m$^{-2}$ during "+T +IWV" periods relative to normal conditions. The effect of reduction in LW CRE during warm and moist conditions in summer is consistent with findings by Cox et al. (2015) and Ebell et al. (2019).

(8) Moist conditions increase the mean net CRE at the surface in autumn, winter, and spring by 5–37 W m$^{-2}$ with respect to normal conditions. This change is mostly defined by cloud radiative properties in LW, which are related to enhanced cloudiness,





LWP, and IWP. Dry conditions reduce the mean net CRE by 2–37 W m$^{-2}$ in autumn, winter, and spring. In summer the net CRE is dominated by the SW CRE and, therefore, moist conditions show stronger cloud related surface cooling. During dry conditions in summer there is an increase in the mean net CRE by 49 m$^{-2}$.

(9) Long-term radiosonde observations show significant trends in the IWV anomaly occurrence. Moist anomalies are getting more frequent with a slope varying for different seasons from 2.8 to 6.4% decade$^{-1}$, while occurrence of dry anomalies declines at rates from -12.9 to -4% decade$^{-1}$. Similar results were found for Greenland in study by Mattingly et al. (2016), which shows that most pronounced increasing moist and decreasing dry IWV patterns were in winter. Since moist and dry anomalies are associated with the North Atlantic and the North Pacific, respectively, our results are consistent with findings of Mewes and Jacobi (2019). The authors showed an increase and decrease in occurrence of the North-Atlantic and North-Pacific air transports in winter, respectively. Dahlke and Maturilli (2017) also showed an increase in occurrence of air masses coming from North Atlantic in winter season. Matthes et al. (2015) reported that cold spell events are becoming less frequent in winter and summer for the whole Arctic region.

(10) Since the anomalies are related to a certain patters in cloud properties and CRE, the trends in the anomaly occurrence over the past 25 years may have lead to increased cloud occurrence, LWP, IWP and, therefore, to higher cloud related surface warming. In addition, if the trends of anomaly occurrences persist in the future, CRE might be more positive in autumn, winter, and spring and more negative summer.

These results show some aspects on how large-scale air transportation may influence atmospheric conditions and, consequently, CRE at Ny-Ålesund. This information is essential for better understanding of relations between these three components of the Arctic climate. As we indicated in this study, the significant trends in the occurrence of anomalous conditions are expected to lead to changes in cloud properties and their radiative effect. Nevertheless, qualitative estimates of these changes are challenging since a long-term Cloudnet dataset at Ny-Ålesund is not currently available. Within the AC$^3$ (ArctiC Amplification: Climate Relevant Atmospheric and SurfaCe Processes, and Feedback Mechanisms) project the cloud measurements are planned to be continued.

*Data availability.* The radiosonde data were taken from the information system PANGAEA: https://doi.org/.10.1594/PANGAEA.845373 (Maturilli and Kayser, 2016), https://doi.org/10.1594/PANGAEA.875196 (Maturilli and Kayser, 2017b), https://doi.org/10.1594/PANGAEA.879767 (Maturilli, 2017a), https://doi.org/10.1594/PANGAEA.879820 (Maturilli, 2017b), https://doi.org/10.1594/PANGAEA.879822 (Maturilli, 2017c), and https://doi.org/10.1594/PANGAEA.879823 (Maturilli, 2017d). The Cloudnet data are available at the Cloudnet website (http://devcloudnet.fmi.fi/). The FLEXTRA data are available at the nilu website (https://projects.nilu.no//ccc/trajectories/evdc/). The HATPRO MWR data is available in the website of the information system PANGAEA: https://doi.pangaea.de/10.1594/PANGAEA.902140 (Nomokonova et al., 2019a), https://doi.pangaea.de/10.1594/PANGAEA.902142 (Nomokonova et al., 2019b), https://doi.pangaea.de/10.1594/PANGAEA.902143 (Nomokonova et al., 2019c), https://doi.pangaea.de/10.1594/PANGAEA.902096 (Nomokonova et al., 2019d), https://doi.pangaea.de/10.1594/PANGAEA.902098 (Nomokonova et al., 2019e), https://doi.pangaea.de/10.1594/PANGAEA.902099 (Nomokonova et al., 2019f), https://doi.pangaea.de/10.1594/PANGAEA.9 (Nomokonova et al., 2019g), https://doi.pangaea.de/10.1594/PANGAEA.902146 (Nomokonova et al., 2019h), https://doi.pangaea.de/10.1594/PANGAEA. (Nomokonova et al., 2019i).



*Author contributions.* TN applied the statistical algorithm, performed the analysis, prepared and wrote the manuscript. KE, UL, MM contributed with research supervision, discussions of the results and manuscript review. KE applied the RRTMG for Ny-Ålesund to derive vertically resolved SW and LW fluxes. MM provided long-term radiosonde dataset. CR provided instrumentation data for this study.

525 *Competing interests.* The authors declare that they have no conflict of interest.

*Acknowledgements.* We gratefully acknowledge the funding by the Deutsche Forschungsgemeinschaft (DFG, German Research Foundation) – Project Number 268020496 – TRR 172, within the Transregional Collaborative Research Center "ArctiC Amplification: Climate Relevant Atmospheric and SurfaCe Processes, and Feedback Mechanisms (AC)[3] in sub-project E02. We acknowledge the team of the AWIPEV Research Base in Ny-Ålesund for helping us in operating the cloud radar, MWR, ceilometer and launching radiosondes. We gratefully acknowl-
530 edge the Aerosol, Clouds, and Trace gases Research Infrastructure (ACTRIS) and particularly Ewan O'Connor for the application of the Cloudnet algorithm to the Ny-Ålesund dataset. NILU is acknowledged for providing the FLEXTRA trajectories (www.nilu.no/trajectories) used in this study.





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





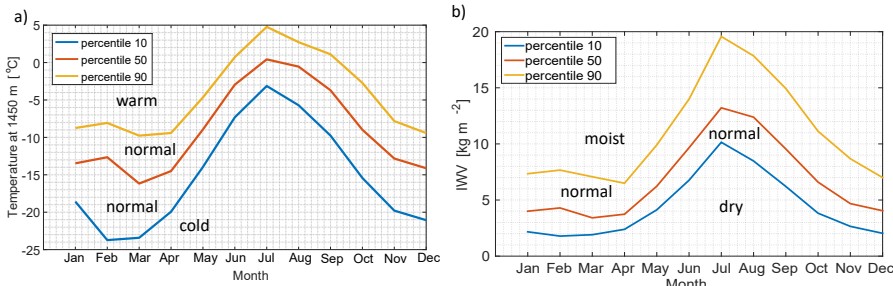

**Figure 1.** Monthly percentiles of 6-hourly (a) mean temperature at 1450 m and (b) IWV from microwave radiometer at Ny-Ålesund from 2011 to 2018 used as criteria for determination of periods of decreased (below 10th percentile, blue line) and increased (above 90th percentile, yellow line) T and IWV. Red line corresponds to the 50th percentile.



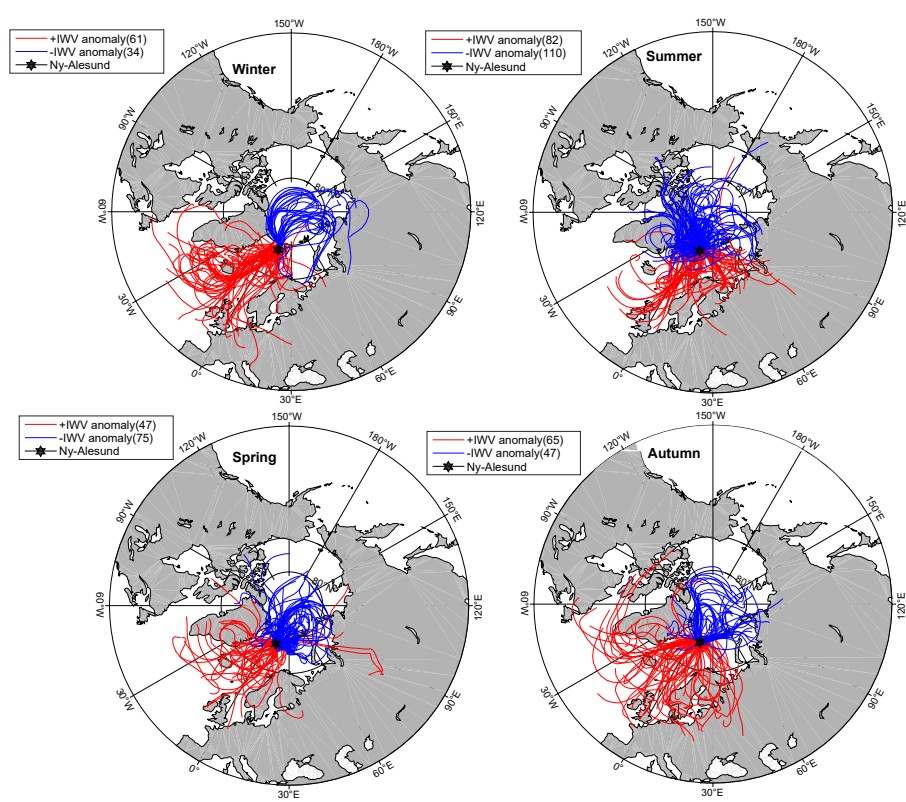

**Figure 2.** Backward trajectories (6 days) for the periods of +IWV and -IWV anomalies arriving at Ny-Ålesund at 1500 m from June 2016 to October 2018. The black star shows the location of Ny-Ålesund. Numbers in brackets show the number of back trajectories available for the corresponding anomaly class. Note that the numbers might be different from those provided in Table 1 due to the lower availability of the back trajectories pathways.



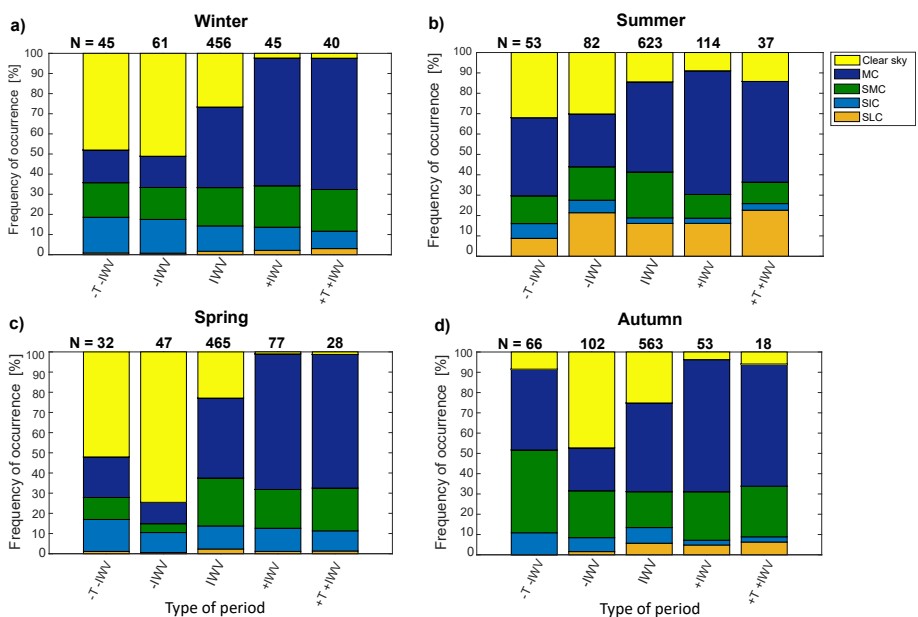

**Figure 3.** Frequency of occurrence of different cloud types for different anomaly periods for winter(a), summer(b), spring(c), and autumn (d). The frequency is normalized to the total number of cases of each anomaly type period. Numbers at the top of bars for each anomaly type show the number of periods included in the corresponding anomaly type based on 6-hourly mean IWV and 1450 m temperature. MC denotes multi-layer clouds, SMC, SIC and SLC stand for single-layer mixed-phase, ice, and liquid clouds, respectively.



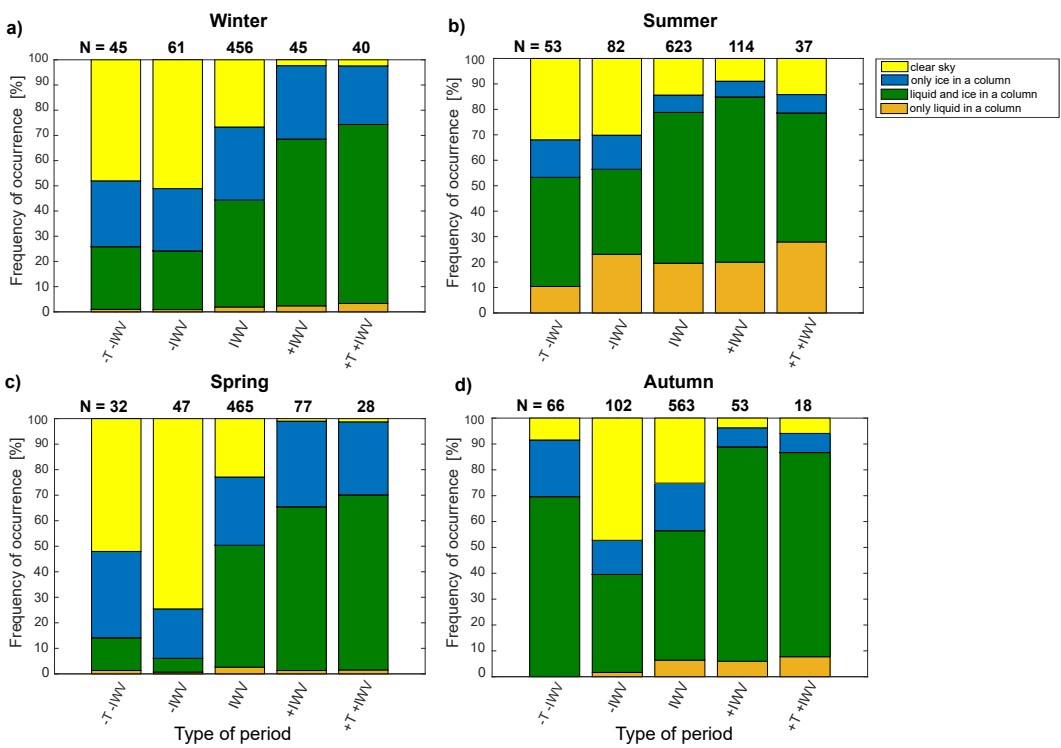

**Figure 4.** Frequency of occurrence of different types of hydrometeors during different anomaly periods for winter(a), summer(b), spring(c), and autumn (d). Numbers at the top of bars for each anomaly type show the number of periods included in the corresponding anomaly type based on 6-hourly mean IWV and 1450 m temperature.



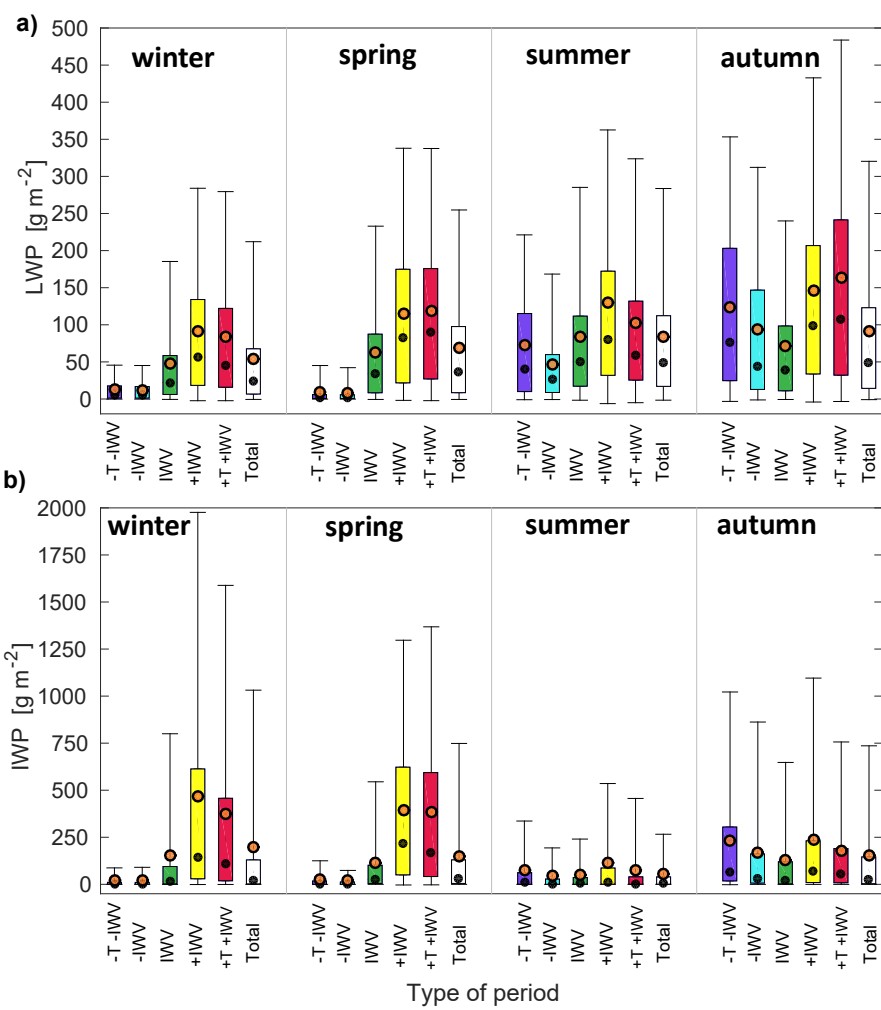

**Figure 5.** LWP (a) and IWP (b) for different anomaly periods and different seasons. Boxes indicate the 25th and 75th percentiles. Upper and lower whiskers show the 95th and 5th percentiles. The white boxes include all cases within a season. Mean (median) values are shown by the orange in black (black) circle marker.

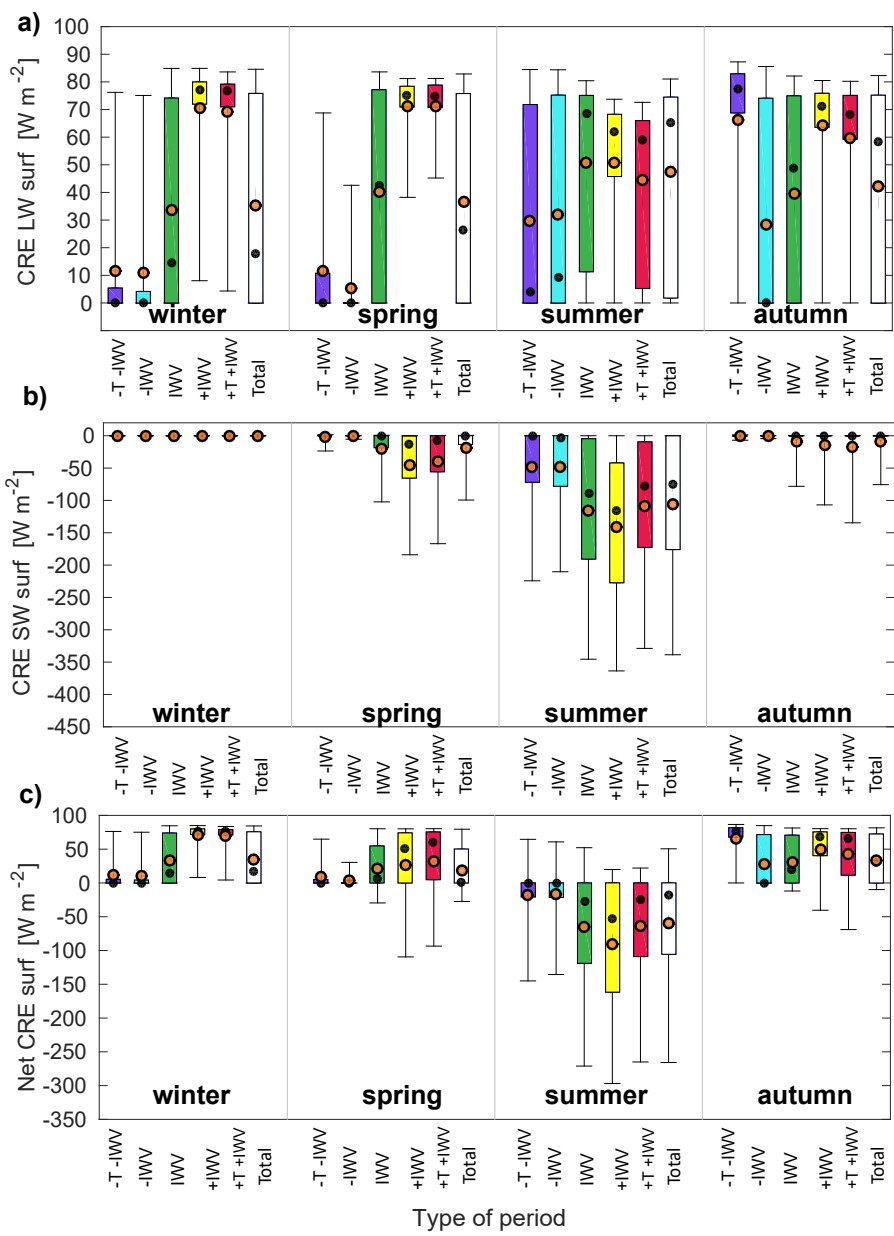

**Figure 6.** SW (a), LW (b), and net cloud radiative effect (c) at the surface for different anomaly periods and different seasons. Boxes indicate the 25th and 75th percentiles. Upper and lower whiskers show the 95th and 5th percentiles. The white boxes include all cases within a season. Mean (median) values are shown by the orange in black (black) circle marker.

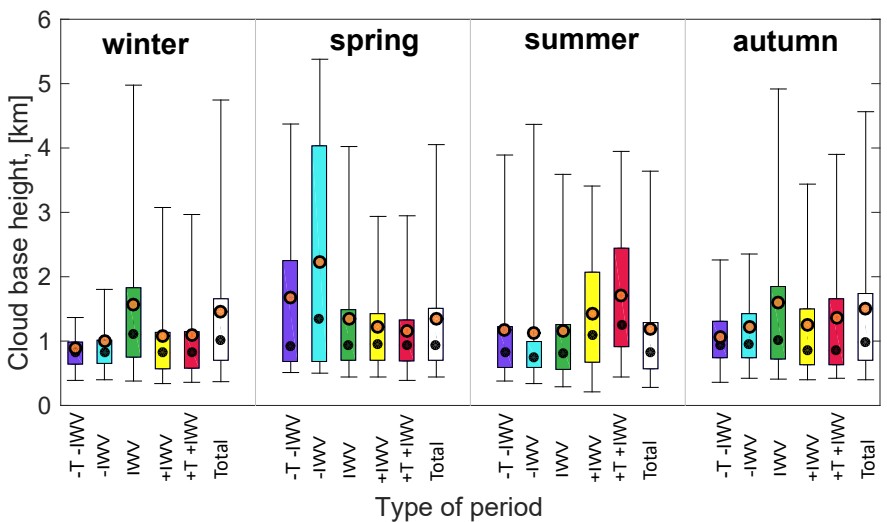

**Figure 7.** Cloud base height of liquid-containing clouds for different anomaly periods and different seasons. Boxes indicate the 25th and 75th percentiles. Upper and lower whiskers show the 95th and 5th percentiles. The white boxes include all cases within a season. Mean (median) values are shown by the orange in black (black) circle marker.

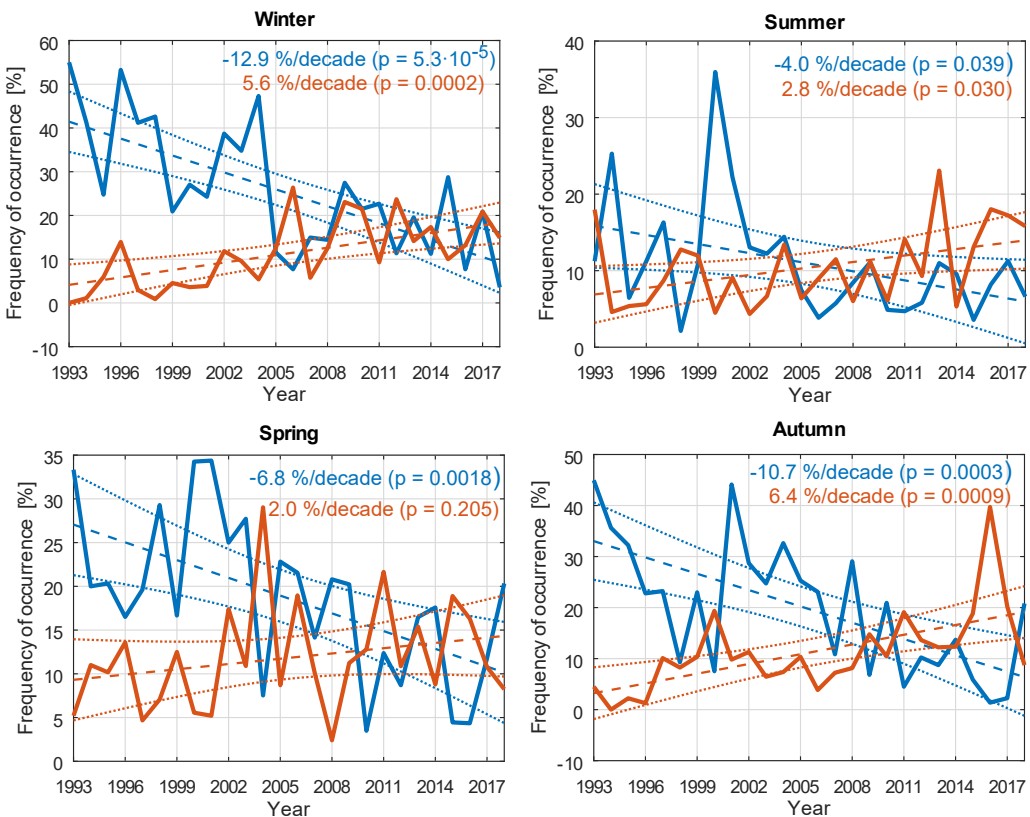

**Figure 8.** Frequency of occurrence of moist ("+IWV", red line) and dry ("-IWV", blue line) events using radiosonde data from 1993-2018 for different seasons (winter (a), summer (b), spring (c), autumn (d)). The red and blue lines correspond to the frequency of occurrence of dry and moist events, respectively. The dashed lines show the linear trend. Dotted lines show the 95th confident intervals for the trends derived by the bootstrapping resampling method. The significance (p value) of the two-sided t-test at 95% level is shown in brackets.





**Table 1.** Number of 6-hourly long periods with increased ("+IWV") and decreased ("-IWV") IWV for the whole period of cloud observations from 2016 to 2018 and for different seasons. Cases with both, increased and decreased T and IWV ("+T +IWV" and "-T -IWV") are also shown. "IWV" corresponds to periods with normal IWV values, regardless which T class the period has. The % values are with respect to all 6-hourly long periods included in the study. See text for more details.

| Type of period | Winter, n cases (%) | Spring, n cases (%) | Summer, n cases (%) | Autumn, n cases (%) | all seasons, n cases (%) |
|---|---|---|---|---|---|
| +T +IWV | 45 (8.0) | 32 (5.4) | 53 (6.5) | 66 (9.2) | 196 (7.3) |
| +IWV | 61 (10.9) | 47 (8.0) | 82 (10.0) | 102 (14.2) | 292 (10.9) |
| IWV | 456 (81.1) | 465 (78.9) | 623 (76.1) | 563 (76) | 2107 (78.4) |
| -IWV | 45 (8.0) | 77 (13.1) | 114 (13.9) | 53 (7.4) | 289 (10.7) |
| -T -IWV | 40 (7.1) | 28 (4.8) | 37 (4.5) | 18 (2.5) | 123 (4.6) |



**Table 2.** Absolute and relative changes in $CRE_{SW}$, $nCRE_{SW}$, and surface albedo ($\alpha$) related to dry and moist anomalies. The absolute changes are calculated as a difference between anomalous and normal cases. The relative changes are shown in brackets and are given in percent with respect to normal conditions. Mean values of $CRE_{SW}$ (in W m$^{-2}$), $nCRE_{SW}$, and $\alpha$ during normal condition are shown in the rightmost column "Normal conditions.

| Parameter | "–IWV" | | | "+IWV" | | | Normal conditions | | |
|---|---|---|---|---|---|---|---|---|---|
| | $\Delta CRE_{SW}$ | $\Delta nCRE_{SW}$ | $\Delta\alpha$ | $\Delta CRE_{SW}$ | $\Delta nCRE_{SW}$ | $\Delta\alpha$ | $CRE_{SW}$ | $nCRE_{SW}$ | $\alpha$ |
| Spring | +18.8(–95) | +0.2(–98) | +0.05(+7) | –25.2(+128) | –0.3(+156) | –0.06(–7) | -19.73 | -0.31 | 0.81 |
| Summer | +67.2(–58) | +0.2(–51) | +0.03(+26) | –25.6(+22) | –0.1(+19) | +0.06(+50) | -115.71 | -0.59 | 0.13 |
| Autumn | +8.8(–94) | +0.2(–60) | +0.45(+173) | –4.9(+52) | –0.4(+115) | –0.08(–30) | -9.39 | -0.31 | 0.26 |