# Peer review of "The influence of water vapor anomalies on clouds and their radiative effect at Ny-Ålesund"

_Atmospheric Chemistry and Physics, 2019_

## Referee Comment (RC1) · Anonymous Referee #1 · 12 Nov 2019

The manuscript "The influence of anomalous atmospheric conditions at Ny-Ålesund on clouds and their radiative effect" addresses how moisture and temperature anomalies are related to cloud properties and cloud radiative effect at Ny-Ålesund in the Arctic. First, origin of the anomalies was studied based on trajectory analyses. Then, clouds and their radiative properties were analyzed in those anomalous conditions. Finally, the trends in occurrence of dry and moist anomalies were analyzed based on 25 years of radiosounding data from Ny-ålesund. The study presents valuable scientific results, the idea of the study is good and the analyses fit logically together. However, the manuscript, in its current form, does not manage to give a clear picture of the study and its main findings. The text is heavy to read and it is not everywhere logically

structured. Before the study can be published, the structure and text need careful, substantial revision.

General comments:

- According to the title, the study addresses "anomalous atmospheric conditions", which could basically be anything. I recommend to have a more explicit title (for example "The influence of water vapor and temperature anomalies on clouds and their radiative effect at Ny-Ålesund").

- The text is very long and heavy to read. In order to help the reader to see the value and main results of the study the text needs substantial revision. I see that the manuscript would benefit from a notable cut in length (even cut of 1/3 of the length) to make its main content more clear.

- The structure is not clear. Parts belonging to introduction are found in "Results" and "Conclusions". Some of the results are already presented in "Methods". Methods and data are presented in an order, which is not logical.

- The atmospheric circulation behind the trajectories is described at an overly simplified level. For example, the authors write several times about "air circulations in the Arctic region" as a source of dry anomalies. These circulations, and their dynamical setting, need to be more precisely described.

- Ny-Ålesund is largely affected by the orography, but the orographical effects are not addressed in this study. Even if the impacts of orography are not the main focus here, they cannot be neglected. How representative are the results of this study for the Arctic? Do they only represent relations seen at Ny-Ålesund, or do the results represent the Arctic conditions more generally? Orography has large impacts on the cloud formation, and therefore it should matter whether the flow meets mountains before it arrives to Ny-Ålesund.

- Introduction covers many relevant topics, but is rather scattered. It would benefit from

a clearer focus.

- What are the accuracies (uncertainties) of the different instruments and data? Please give information.

- In Section 4.5, I recommend to show a comparison of radiosonde IWV and Microwave radiometer IWV for the overlapping period to show how well they agree.

- In section 4.5, impacts of sea ice retreat around the archipelago of Svalbard are not considered at all. The changes in sea ice have, for sure, affected the moisture conditions in Ny-Ålesund and they should be addressed.

- "Summary and conclusions" should be half of its current length to emphasize the MAIN (and thus not all) results of the study. Please state the main findings here; clearly and relatively shortly. My recommendation is to summarize the results for each of the anomaly type (IWV+, IWV-...) instead of going through all the variables separately as in the results section. This would nicely summarize the impacts of the anomalies, which are the main focus of the study (as also indicated in the title).

Specific comments:

- Lines 12-13: The analyses of past trends does not say anything about the future. Therefore, please use the past tense here. "have become", "have increased".

- Line 12: add "ranging" before "from -12.8..."

- Lines 13-16: The two last sentences are not understandable

- Lines 47: Clarify which differences are given here, because it is difficult to understand. What are these values?

- Line 62: "the specific synoptic regime" is a vague expression.

- Line 69: temperature at which level?

- Line 73: Add "in Ny-Ålesund"

- Lines 73-85: This part should be condensed and the text should have a more logical flow.

- Lines 101-103: This project information is unnecessary. It is enough to mention the project in the acknowledgements.

- Line 136: "were analyzed" instead of "will be analyzed"

- Line 138: Add some reference to the model.

- Line 151: Is this operational forecast or reanalysis data? Specify.

- Section 2.6 should be placed after other measurements (to be 2.4.) And then the methods should follow.

- Section 3 could be much shorter. There is a lot of repetition and many things could be expressed in a shorter way.

- Section 2: Define months of winter, spring, summer and autumn somewhere.

- Line 181: Add "horizontal" before "transport"

- Line 182: "Coming from lower or higher latitudes" means basically from anywhere. This is too vague.

- Lines 186-188: The citation to Graversen (2006) is a bit unconnected here.

- Lines 190-192: Again, description of circulation is vague.

- Lines 192-193: I cannot see that the MOST of dry anomalies in summer are coming with the air from Canada and Greenland in Fig. 2.

- Line 193: pathways for dry or moist air? Specify.

- Line 193: I do not see the two distinct pathways from south-east and west!

- Lines 195: Is this statistically significant. If not tested, use another word than "significant". Give percentage value here.

- Results: The section 3 includes already results, so results section cannot start here.

- Lines 211-214: This is actually about methods, and could be omitted.

- Line 218: How much lower? Specify.

- Lines 228-229: I disagree. These results are mentioned many times in this manuscript, without any deeper understanding. Please investigate this event so that you can say something what was so special in it.

- I cannot find any reference to Fig. 3 in the text.

- Lines 237-241: This is introduction again. Omit or move.

- Lines 244-245: What is meant by "only in summer" here? Is 8% little or much?

- Line 275: The sentence starting "In contrast,..." is not clear.

- Lines 277-289: A lot is written about aerosols here without a concrete connection to this study. Consider omitting or make the link to this study clearer.

- Lines 286-289: Remove until "Figure 6 summarizes..." to remove repetition and introduction-type text.

- Line 291: Where and when? Is this a new result or based on a previous study which is mentioned.

- Line 291: Add unit.

- Line 298: "increase in mean LW CRE" Which "mean"?

- Line 300: I do not see the lower LW CRE during moist anomalies in Fig. 6.

- Lines 300-305: Effects of relative humidity remain unclear based on this. Explain why relative humidity can affect?

- Lines 312-313: In addition, the summer cloud often radiate LW as a black-body, so an increase in LWP/IWP will not much affect their LW radiation in summer.

- Lines 328-330: Do the author say that if LWP and IWP and frequency of occurrence of clouds do not vary, the cloud properties cannot vary? What about droplet size and aerosols affecting the SW CRE?

- Lines 344-347: Omit or shorten.

- Line 408: Are the percentile threshold values taken here from the Microwave radiometer data or radiosounding data?

- Lines 422-426: remove from here, because this part belongs to " Summary and conclusions".

- Line 440: Was the correlation analyzed? If not, do not use the word "correlate".

- Lines 442-443: Again, what is specifically meant by "air circulating in the Arctic"?

- Line 445: Statistically significant?

- Line 446: "counterclockwise air circulations" could be "cyclonic".

- Line 447: What is meant by "this type of circulation"?

- Line 449: See my earlier comment about these directions. I could not see these results in the figure.

- Lines 444-450: This part is far too long and the dynamical part is not well enough explained.

- Figure 8: From which level (height) the radiosonde IWV is taken?

---

## Referee Comment (RC2) · Anonymous Referee #2 · 5 Jan 2020

This paper uses 2.5 years of data collected at Ny Alesund to (a) identify the upper 10% and lowest 10% of the cases by integrated water vapor ("wet" and "dry" anomalies, respectively) and also the upper 10% and lower 10% of the cases by temperature at ~1500 m MSL ("warm" and "cold" anomalies, respectively) for each season. The differences in the cloud properties (macro- and microphysical) for the anomalous periods relative to the "other 80%" are shown and discussed. The authors then look at the differences in the cloud radiative forcing at the surface for these anomalous conditions, and correlate them with the differences in cloud properties and IWV.

The paper is reasonably well written, the tables and figures are concise and well pre-

sented, and the references look reasonable. I have some comments / suggestions that I believe should be considered before the paper is accepted for publication. These are outlined below. The main concern I have is associated with the sampling uncertainty; I speak more about this below.

• Why wasn't the cloud radar included in the instrument list / description section? It is key for the CloudNet products, which are key for the rest of the paper. • Why were NWP thermodynamic profiles used in the CloudNet classification, and not the profiles retrieved from the HATPRO? • Line 138: the RRTMG should be referenced, as it is a critical component to this study • Are the uncertainties in the retrieved cloud properties (and in particular the phase classification) important to this study? • What vertical separation is required to identify multi-layer clouds? • Are the backtrajectories allowed to touch the surface? Or if they touch the surface, do you call that the origin point for the trajectory? • Line 168: The HATPRO period is from 2011 to 2018. This isn't a very long period. Using the longer radiosonde record, how representative is the 2011-2018 period? It is clear that the mean values won't be the same (hence the trends over time), but is the range of variability (e.g., standard deviation of the IWV)) the same for the short period as the longer period? • Ditto for the 2016-2018 period. • I have a lot of questions regarding the sampling uncertainty, especially in the 2016-2018 dataset. The authors themselves hinted at this at line 225 when they say "Such a short period of time would probably not be representative. . .". Table 1 shows that the number of cases in these "outlier" categories is small (less than 100, often less than 50). This is, by far, the biggest weakness of the paper. The authors must augment their discussion to talk about sampling uncertainties, which includes the two questions above regarding representativeness. • Line 227: you are talking about the "-T -IWV" cases, and stating that the LWC and IWC are larger than in normal conditions. This is indeed counterintuitive. Perhaps the authors could look at the column RH value (computed as IWV / saturated_IWV, where the temperature profile is retrieved from the HATPRO) to see if there are any differences in this value? It would seem that the column RH in the "-T -IWV" case must be larger than the normal conditions for the cloud

water paths to be larger. . . • Section around line 249, where the discussion focuses on ice clouds: I think that the authors should consider breaking the analysis into high ice clouds (e.g., cirrus) and "boundary layer" ice clouds. Generally speaking, I would not expect the former to have much of a dependence on IWV, whereas I can see how the BL ice clouds could depend on changes in IWV. • Line 266: To state that there is a 2x increase in LWP is not really clear enough. If the LWP is less than 10 g/m2, then a 2x increase is 20 g/m2 which is still close to the uncertainty in the HATPRO LWP retrievals. Would those retrievals have sensitivity to the atmospheric state, or in other words, is this 2x change in LWP an artifact of the retrieval? • Line 269: again, where the ice clouds are located vertically may be important for this statement. • Line 291: need to add units to the "0 to 85" • Line 464: "excess and shortage" are odd words here. I think this phrase must be changed to be more clear • Line 466: "reduction of LWP and IWP by an order of magnitude" seems to suggest both are decreased by a factor of 10, when I believe you only mean the IWP is changed by a factor of 10. • Line 500: "patterns" is misspelled • Fig 6a: Is the autumn "-T -IWV" bar where it is due to that one 5-day period? I think the answer is yes, and this is a great example indicating that the sampling errors must be better discussed. And a note should be made in the caption here. • Fig 6c: it would be nice to have a horizontal line at CRE = 0.

---

## Author Comment (AC1) · 20 Feb 2020

Thank you for constructive comments, which helped to improve the manuscript. Please find our detailed responses in the supplement.

Please also note the supplement to this comment: https://www.atmos-chem-phys-discuss.net/acp-2019-985/acp-2019-985-AC1-supplement.zip

---

## Author Response (AR1)

**Reply to reviewer #1**

We would like to thank the reviewer #1 for the constructive comments, which helped us to improve the manuscript. We have considered all the recommendations. Below, the reviewer's comments are in red. Our replies are given in black. Please note, that in the statements of the reviewer lines and figures refer to the original manuscript and may have changed in the revised version.

**1 General comments**

1) According to the title, the study addresses "anomalous atmospheric conditions", which could basically be anything. I recommend to have a more explicit title (for example "The influence of water vapor and temperature anomalies on clouds and their radiative effect at Ny-Ålesund").

✓ We changed the title to "The influence of water vapor anomalies on clouds and their radiative effect at Ny-Ålesund".

2) The text is very long and heavy to read. In order to help the reader to see the value and main results of the study the text needs substantial revision. I see that the manuscript would benefit from a notable cut in length (even cut of 1/3 of the length) to make its main content more clear.

✓ We implemented several modifications in order to shorten the manuscript a bit and make the content clearer. First, the introduction was shortened. Second, considering comments of the both reviewers, we decided to exclude temperature anomalies ("-T-IWV" and "+T+IWV") from the manuscript. Third, the summary was reorganized and shortened.

✓ Please note, that in order to address some of the reviewer's comments we had to include some additional discussions (e.g. sampling uncertainties, comparison of IWV from radiosondes and MWR). The additional figures were placed in the supplementary material, but the discussion was added to the manuscript.

✓ Overall, the shortening of the manuscript is about 15%. Since the journal does not have any restrictions on the length, we would still like to leave the remaining results in the manuscript.

3) The structure is not clear. Parts belonging to introduction are found in "Results" and "Conclusions". Some of the results are already presented in "Methods". Methods and data are presented in an order, which is not logical.

✓ We agree that there was a misleading structure. We reorganized the structure. Please see the corrections made in the answers to specific comments regarding these issues.

4) The atmospheric circulation behind the trajectories is described at an overly simplified

level. For example, the authors write several times about "air circulations in the Arctic region" as a source of dry anomalies. These circulations, and their dynamical setting, need to be more precisely described.

✓ Please note, that the main idea of this study is to show how the events of dry and moist conditions influence the cloud appearance and their radiative effect at Ny-Ålesund. The analysis of the back trajectories was made in order to check whether our definition of the anomalous conditions is consistent with literature, where a number of studies show that moist conditions at Ny-Ålesunds are caused by air masses coming from the North Atlantic, while dry conditions are typically caused by Air coming from the Arctic regions. In order to check the consistency, we checked primary directions of the air flow related to moist/dry conditions at Ny-Ålesund. Since, these directions agree with the literature we assume that the used definition of the anomalies is valid. We agree that in the initial version of the manuscript this was not explained well enough. We introduced a separate subsection (4.1) on this.

5) Ny-Ålesund is largely affected by the orography, but the orographical effects are not addressed in this study. Even if the impacts of orography are not the main focus here, they cannot be neglected. How representative are the results of this study for the Arctic?
Do they only represent relations seen at Ny-Ålesund, or do the results represent the Arctic conditions more generally? Orography has large impacts on the cloud formation, and therefore it should matter whether the flow meets mountains before it arrives to Ny-Ålesund.

✓ In the current manuscript we analyze relations between IWP and cloud properties in a vertical column at Ny- Ålesund. Of course, the atmosphere at Ny- Ålesund is influenced by a large number of factors including orographic effects, ocean, aerosols, and many others. These factors are explicitly mentioned in the introduction section. Please note, that the fact that we do not analyze these factors in this manuscript does not mean that they are neglected. These factors affect both IWV and cloud properties and thus the effects of these factors are in the measurements. The problem is that it is hard to identify individual effects by the different factors in the observations. Some first steps into this direction (identifying an influence of orography on clouds in Ny- Ålesund have been already made within the (AC)[3] project and are ongoing). The first results have been recently published or accepted by ACP (e.g. Schemann and Ebell 2020, and Gierens et al 2019).

✓ The following sentences were added to the section 4.3: "Since the anomaly type cannot fully explain this effect, it is probably also related to other factors such as differences in aerosol load, impact of local effects due to the surrounding orography, and an influence of the ocean. For instance, the seasonal change in the aerosol type affects the activation ability of CCN and IN efficiency in the Svalbard region which in turns influence the cloud

formation. In addition, the reduction in sea ice around the Svalbard archipelago and on the fjords may lead to more evaporation and therefore, affect cloud conditions. Since Ny-Ålesund is surrounded by mountains up to 800 m, the air flow is influenced by the local orography in the lowest 1 km altitudes (Maturilli and Kayser, 2017a) and thus might have an impact on cloud formation and change cloud properties."

✓ The results obtained within this work are relevant only for Ny-Ålesund and not mapped to the whole Arctic. Therefore, these results should not be considered as general Arctic conditions. Since Ny- Ålesund is located in the warmest part of the Arctic, the results are likely to be different with respect to other Arctic sites. This is one of the motivations why clouds are analyzed at Ny- Ålesund even though long-term cloud observations are available for example for Canadian Arctic and Greenland. This uniqueness of Ny-Ålesund is explicitly mentioned in the introduction. The comparison studies with other Arctic sites are planned in the future when more cloud measurements at Ny-Ålesund will be available.

✓ Please note, that directional dependence of clouds on wind direction at Ny- Ålesund has been recently analyzed by Gierens et al. 2019 (accepted by ACP).

6) Introduction covers many relevant topics, but is rather scattered. It would benefit from a clearer focus.

✓ The introduction has been restructured and shortened.

7) What are the accuracies (uncertainties) of the different instruments and data? Please give information.

✓ The detailed information on different instruments used in this study and their uncertainties has been provided in Nomokonova et al. 2019 that was published in the same special ACP issue. This study is properly referenced in the instrumentation section. The study gives a full overview of the instruments, retrievals, and their uncertainties. In the current manuscript we briefly described the instruments. If we include further information on the instruments the manuscript will become even more longer. In this case we would also publish the same information twice. This is why we would like to keep the section as it is.

8) In Section 4.5, I recommend to show a comparison of radiosonde IWV and Microwave radiometer IWV for the overlapping period to show how well they agree.

✓ Thank you for highlighting this aspect. We compared IWV derived from both instruments for the overlapping period. The results are shown in the figure below:

[Figure]

Figure 1. IWV comparison between MWR (mean value within 15 min after a radiosonde launch) and radiosonde (only around 11 UTC radiosondes included) for the period from 2011 to 2017. The data are from the AWIPEV station in Ny-Ålesund.

✓ This figure was added to the Supplement material (please see the attached file of the Supplement material, Figure S2). We also added a discussion on this comparison in the text of the manuscript: "Values of IWV retrieved from MWR were compared with ones derived from radiosondes for the period from 2011 to 2017 when both observations were available. IWV from MWR was averaged over 15 min interval after radiosonde launch. The results of the comparison are shown in the supplement material (Fig. S1) and are in a good agreement with the root mean squared difference of 0.56 kg m$^{-2}$ and a bias is close to zero." Please see section 2.4.

9) In section 4.5, impacts of sea ice retreat around the archipelago of Svalbard are not considered at all. The changes in sea ice have, for sure, affected the moisture conditions in Ny-Ålesund and they should be addressed.

✓ We agree that the sea ice decline can affect the increased occurrence of moisture event and lead to the reduction of dry events. Thank you for highlighting this aspect. We added the discussion on this topic.

✓ Similar to the orographic effects (see comment 5 above), the influence of the sea ice extent (and any other effect) on water vapor and clouds is captured by measurements but it is hard to quantify this effect separately from other factors.

✓ We added the following discussion at the end of the Section 4.5: "In addition to the influence of air mass transport toward Svalbard region changes in sea-ice coverage around the archipelago of Svalbard might also impact the occurrence of moist and dry events. During summer and autumn the sea ice coverage is the lowest which can lead to enhanced evaporation and latent heat exchange between the ocean and the Arctic atmosphere. However, the largest sea ice loss rate over Svalbard has been observed in winter (Isaksen et al., 2016). Beside the changes of the sea ice coverage around the Svalbard archipelago a reduction of the local fjord ice cover has an impact on the local

climate (Isaksen et al., 2016, Dahlke et al., 2020) and therefore, might also lead to a change in occurrence of anomalous atmospheric conditions at Ny-Ålesund."

✓ The reference information of Isaksen et al., 2016, Dahlke et al., 2020 were added to the reference list.

10) "Summary and conclusions" should be half of its current length to emphasize the MAIN (and thus not all) results of the study. Please state the main findings here; clearly and relatively shortly. My recommendation is to summarize the results for each of the anomaly type (IWV+, IWV-: : :) instead of going through all the variables separately as in the results section. This would nicely summarize the impacts of the anomalies, which are the main focus of the study (as also indicated in the title).

✓ Thank you for highlighting to this aspect and recommendations. We made changes according to the recommendations.

✓ The Section 5 "Summary and conclusions" was reorganized in order to combine the results of this study into two groups related to anomaly types "+IWV" and "-IWV", respectively. The changes also covered some sentences which were rephrased and removed from the section.

**2 Specific comments:**

1) Lines 12-13: The analyses of past trends does not say anything about the future. Therefore, please use the past tense here. "have become", "have increased".
✓ Corrected

2) Line 12: add "ranging" before "from -12.8 …"
✓ Corrected

3) Lines 13-16: The two last sentences are not understandable
✓ These sentences have been removed

4) Lines 47: Clarify which differences are given here, because it is difficult to understand. What are these values?
✓ Due to the shortening of the introduction this part has been removed from the manuscript.

5) Line 62: "the specific synoptic regime" is a vague expression.
✓ We rephrased the following sentence: "A number of studies focus on observations at Ny-Ålesund located in the Svalbard region (Wendisch et al., 2019; Maturilli et al., 2013; Maturilli and Ebell, 2018; Yeo et al., 2018), an Arctic area where air masses transported from the lower latitudes bring more moisture in comparison to the rest of the Arctic (Dahlke and Maturilli 2017; Mewes and Jacobi, 2019).

6) Line 69: temperature at which level?
✓ Due to the shortening of the introduction this part has been removed from the manuscript.

7) Line 73: Add "in Ny-Ålesund"
✓ Added.

8) Lines 73-85: This part should be condensed and the text should have a more logical flow.
✓ We made this part shorter and added the connections between the sentences. The following sentences were rewritten: Dahlke and Maturilli (2017) showed an increasing air mass transport through the North Atlantic pathway and reducing flow from the north in the winter season in Ny-Ålesund. Yeo et al. (2018) investigated how the advection of warm and cold air masses affects cloudiness, longwave fluxes at the surface and near-surface temperature at Ny-Ålesund during winter. The authors analyzed a 10-day period in February with alternating warm and cold conditions related to distinct circulation patterns. During cold periods Yeo et al. (2018) observed a reduced cloudiness and downwelling longwave flux of 200–230 W m^2. In contrast, warm periods were associated with cloud occurrence close to 100 % and enhanced downwelling longwave flux of 300 W m^2. Since the author studied only a short period, an analysis of longer cloud observations is still needed.

9) Lines 101-103: This project information is unnecessary. It is enough to mention the project in the acknowledgements.
✓ (AC)³ is a huge German initiative that we would like to emphasize and keep it here as well.

✓ We made this part shorter: "Within the Transregional Collaborative Research Center TRR 172 on Arctic Amplification ((AC)³; Wendisch et al., 2017), the instrumentation at AWIPEV was complemented with a Doppler cloud radar in June 2016."

1) Line 136: "were analyzed" instead of "will be analyzed"
✓ Corrected

2) Line 138: Add some reference to the model.
✓ We added the references to Mlawer et al., 1997 and Barker et al., 2003 after the RTTMG.

3) Line 151: Is this operational forecast or reanalysis data? Specify.
✓ The meteorological data used as input for FLEXTRA is not from reanalysis it is from initialized operational analyses of the NWP model of ECMWF with a temporal resolution of 6 hours (at 0, 6, 12, and 18 UTC) and a spatial resolution of 1.125 degree.
✓ We rephrased the following sentence: "The calculations of the trajectories are based on operational analysis data from NWP model of the European Centre for Medium range Weather Forecast (ECMWF) with the initialized analyses every 6 hours (at 0, 6, 12, and 18 UTC) …".

4) Section 2.6 should be placed after other measurements (to be 2.4.) And then the methods should follow.

✓ Done

5) Section 3 could be much shorter. There is a lot of repetition and many things could be expressed in a shorter way.

✓ The Section 3 was shortened and split into two parts. We decided to leave only the first paragraph in the Section 3. The remaining paragraphs were moved to a new subsection 4.1 "Consistency check of the defined anomaly periods with existing studies".

6) Section 2: Define months of winter, spring, summer and autumn somewhere.

✓ Please find the months in the section 4.1: "In winter (December, January and February) and spring (March, April and May), dry air typically comes from North of Russia over the North Pole region and northern Greenland. Mewes and Jacobi, (2019) have shown that a similar horizontal air mass transport happens when air from the North Pacific flows into the Arctic. In summer (June, July and August), dry anomalies are mostly associated with air coming from northern Canada and Greenland. In autumn (September, October and November), …"

7) Line 181: Add "horizontal" before "transport"
✓ Added

8) Line 182: "Coming from lower or higher latitudes" means basically from anywhere. This is too vague.
✓ We reformulated: A number of studies have shown that positive and negative anomalies in IWV often result from horizontal transport of air masses from mid- and Arctic latitudes, respectively.
9) Lines 186-188: The citation to Graversen (2006) is a bit unconnected here.
✓ The sentence was removed.

10) Lines 190-192: Again, description of circulation is vague.
✓ We rephrased the following sentences: "In winter (December, January and February) and spring (March, April and May), dry air typically comes from North of Russia over the North Pole region and northern Greenland. Mewes and Jacobi (2019) have shown that a similar horizontal air mass transport happens when air from the North Pacific flows into the Arctic."

11) Lines 192-193: I cannot see that the MOST of dry anomalies in summer are coming with the air from Canada and Greenland in Fig. 2.
✓ We rephrased the following sentence: "In summer (June, July and August), dry anomalies are associated with air coming from the northern part of Canada and Greenland."

12) Line 193: pathways for dry or moist air? Specify.
✓ Modified: "In autumn (September, October and November) during dry anomalies air masses come from the areas north of Greenland and Russia."

13) Line 193: I do not see the two distinct pathways from south-east and west!

✓ Modified: "In autumn (September, October and November) during dry anomalies air masses come from the areas north of Greenland and Russia."

14) Lines 195: Is this statistically significant. If not tested, use another word than "significant". Give percentage value here.

✓ We replaced the word "significant" to "large".

15) Results: The section 3 includes already results, so results section cannot start here.

✓ The Section 3 was shortened and split into two parts. We decided to leave only the first paragraph in the Section 3. The remaining paragraphs were moved to a new subsection 4.1 "Consistency check of the defined anomaly periods with existing studies".

16) Lines 211-214: This is actually about methods, and could be omitted.

✓ The introduction to the section 4 was removed. One sentence was added to the subsection 4.2 "In this section we show how the anomalous conditions are related to cloud occurrence."

17) Line 218: How much lower? Specify.

✓ The expression "is in general lower and …" was deleted to avoid misleading statement. In the same sentence the range of the values for different seasons is provided "..the FOC of clouds ranges from 26 % in spring to 70 % in summer".

18) Lines 228-229: I disagree. These results are mentioned many times in this manuscript, without any deeper understanding. Please investigate this event so that you can say something what was so special in it.

✓ In order to make the manuscript shorter and due to the low number of cases for the temperature anomaly periods (comments by the reviewer 2) we decided to exclude them from the revised manuscript.

✓ However, to answer the questions from reviewers related to "-T-IWV" period, since this period was particularly interesting, we provided a detail analysis here in the replies. Please note, that this description will not be added to the manuscript. We investigated "-T-IWV" in more details and information on this event is provided below. During the 5-day "-T-IWV" period (from 26.09.2018 to 30.09.2018) we observed an increased cloud occurrence (>90%) and higher mean LWP (~120 g m-2) and IWP (~250 g m-2). The figure 2 below shows height-time cross sections of clouds for two first days of the "-T-IWV" period. On 26 Sept 2019 the liquid layer of mixed-phase clouds is located at the top of the cloud. In the afternoon the liquid layer is distributed within cloud boundaries. Measurements of the Doppler velocity from the cloud radar 94~GHz indicate the presence of up and down drafts (not shown).

[Figure]

Figure 2. Cloudnet target classification on 26 and 27 September 2018. The quicklooks are taken from the official Cloudnet website (http://devcloudnet.fmi.fi/ ).

The analyses of radiosonde observations in figures 3-5 below show that on 25092018 (one day before "-T-IWV") there was a temperature inversion, relative humidity with respect to water was ranging from 90-100 % and the wind was blowing from the north-west with higher wind speed in the lowest 1 km. The first day of "-T-IWV" (26092018) was associated with a fast decrease of temperature above 1 km. The wind speed above 1 km increase as well. These cold and windy conditions stayed for the next 5 days. The direction of wind from radiosondes is consistent with the backward trajectories for "-T-IWV" anomaly period (Fig.6). These cold conditions were associated with low pressure system above Barents sea and air masses coming from the northern Greenland which are shown on the screenshots from the online source of the automatic global ICON on the website https://www.ventusky.com/ (Fig.7). One day before the "-T-IWV" period the low-pressure system was only above the Spitzbergen and to the north relative to the archipelago. The screenshots with the air temperature at 850 hPa from the global ICON show that relatively cold air mases passed by the western part of Svalbard on 25092018 while the air masses coming to the Svalbard region on the next days led to the observed decrease in temperature at Ny-Ålesund. The analysis of potential and equivalent potential temperature for 5 days (fig. 9) reveals that during "-T-IWV" there were unstable conditions to vertical motions which were indicated by the decrease of equivalent potential temperature (solid lines) with a height up to 2 km. This led to a convection and therefore, mixing of cold air coming from the north-west reaching Ny-Ålesund with relatively warm and moist air parcels at the surface. This vertical mixing probably promoted enhanced cloud formation. Such complicated conditions are likely to be misinterpreted by the proposed classification scheme.

[Figure]

Figure 3. Temperature, relative humidity, wind speed, and wind direction profiles from radiosonde for 25092018 (one day before the "-T-IWV" period) launched at around 5, 12, 17 and 23 hours (UTC).

[Figure]

Figure 4. The same as Figure 3 but for 26092018 (first day of "-T-IWV" period).

[Figure]

Figure 5. The same as Figure 3 but for 27092018 (second day of "-T-IWV" period).

[Figure]

Figure 6. Backward trajectories for the "-T-IWV" period shown by blue lines.

[Figure]

[Figure]

[Figure]

Air pressure

[Figure]

[Figure]

Figure 7. Screenshots of air pressure from the Global ICON model (The screenshots are taken from the available online source https://www.ventusky.com/?p=71;-27;2&l=temperature-850hpa&t=20180927/2100).

[Figure]

Temperature at 850 hPa, ~ 1500 m

[Figure]

[Figure]

Figure 8. The same as Figure 7 but for temperature at 850 hPa.

[Figure]

Figure 9. Potential (dashed lines) and equivalent potential temperatures (solid lines) obtained from radiosonde observations.

✓ We added the following sentence at the end of the Subsection 3.1 "Identification of periods…" in the revised version of the manuscript: "Since the number of cases for temperature anomalies ("+T +IWV" and "-T -IWV") is relatively low these types of anomalies were not considered for the further analysis.".

✓ The categories "-T-IWV" and "+T+IWV" were removed from the table 1 and the expressions "(not shown)" were added to the following sentences: "Periods with "+T +IWV" anomaly take a major part (about 67%) of all moist anomalous cases (not shown). In contrast, occurrence of "-T -IWV" periods is only about 35% of all dry anomalies in all seasons except winter, when the occurrence is almost 90% (not shown)."

19) I cannot find any reference to Fig. 3 in the text.
✓ Please find the reference to Figure 3 in the second sentence of the subsection 4.2 of the revised manuscript.

20) Lines 237-241: This is introduction again. Omit or move.

✓ This part was removed and the section 3.3 was combined with the previous subsection. Now the combined subsection 3.2 has a title: "Cloud occurrence and phase".

✓ The following sentence replaced the removed part: "Since the phase composition of clouds affects SW and LW radiative properties of clouds Ebell et al. (2020), we also analyzed the occurrence of different types of hydrometeors in the atmospheric column (Fig. 4).

21) Lines 244-245: What is meant by "only in summer" here? Is 8% little or much?
✓ We rephrased the following sentence: "In summer the increase in FOC of liquid containing profiles between moist and normal periods is less pronounced (8%)."

22) Line 275: The sentence starting "In contrast,: : :" is not clear.
✓ Rephrased: "In autumn during wet conditions the increase in mean IWP is about factor of 2 and during dry conditions the mean IWP does not decrease."

23) Lines 277-289: A lot is written about aerosols here without a concrete connection to this study. Consider omitting or make the link to this study clearer.
✓ We removed the long discussion related to aerosols. We added the discussion on different factors and rephrased the following sentences in subsection 4.3.: "Since the anomaly type cannot fully explain this effect, it is probably also related to other factors such as differences in aerosol load, impact of local effects due to the surrounding orography, and an influence of the ocean. For instance, the seasonal change in the aerosol type affects the activation ability of CCN and IN efficiency in the Svalbard region which in turns influence the cloud formation. In addition, the reduction in sea ice around the Svalbard archipelago and on the fjords may lead to more evaporation and therefore, affect cloud conditions. Since Ny-Ålesund is surrounded by mountains up to 800 m, the air flow is influenced by the local orography in the lowest 1 km altitudes (Maturilli and Kayser et al.,2017a) and thus might have an impact on cloud formation and on cloud properties. Note, that within this study the influence of local effects and aerosols is not analyzed."

24) Lines 286-289: Remove until "Figure 6 summarizes: : :" to remove repetition and

introduction-type text.
- ✓ This part was related to the motivation of the current subsection 4.4. And this part leads the reader to smoothly follow the topic of the current section and its connection to the previous sections.
- ✓ We made this part shorter. We left and rephrased the following sentence: "In the previous sections, we showed that water vapor anomalies affect cloudiness and the amount of liquid and ice in a column which will also influence the CRE at the surface at Ny-Ålesund."

25) Line 291: Where and when? Is this a new result or based on a previous study which is mentioned.
- ✓ You are right. This information should be specified in order to avoid misunderstanding. We added a necessary information to the following sentence: "Values of LW CRE at Ny-Ålesund from June 2016 to October 2018 are in the range from 0 to 85 W m$^{-2}$ and agree with the cloud occurrence and amount of liquid in a column (Ebell et al., 2020)."

26) Line 291: Add unit.
- ✓ The units were missing. We added the units.

27) Line 298: "increase in mean LW CRE" Which "mean"?
- ✓ In the previous version it was not clear what periods exhibit the changes in mean LW CRE. We rephrased the following sentence: "This increase in mean LW CRE due to moist anomalies in winter, spring, and autumn is associated…".

28) Line 300: I do not see the lower LW CRE during moist anomalies in Fig. 6.
- ✓ Thank you for highlighting this mistake. This sentence was not correct. We rephrased the following sentence: "In contrast to other seasons, in summer the mean LW CRE during moist anomalies is not higher than under normal conditions."

29) Lines 300-305: Effects of relative humidity remain unclear based on this. Explain why relative humidity can affect?
- ✓ The conditions with increased relative humidity are associated with concurrent increase in amount of atmospheric water vapor. Cox et al. (2015) showed that LW CRE depends on amount of IWV. The authors demonstrated that at the constant temperature and increased water vapor the LW CRE decreases (Fig. 4c, Cox et al., 2015). The LW CRE at the surface is lower for these cases because more absorption and emission from water vapor in the atmosphere between clouds and the surface leads to less emission from clouds transmitted through the atmosphere with increased amount of water vapor. Therefore, lower LW CRE was found for cases with increased humidity and temperature.

  We included an explanation and added the following sentences: "The authors analyzed data from radiative transfer simulations and observations obtained at Barrow, Summit and Eureka. They found that the increase in amount of atmospheric water vapor particularly between a cloud and the surface diminishes the impact of the infrared radiance emitted from the cloud due to more absorption and emission by water vapor

itself. They showed that at constant temperature for higher relative humidity LW CRE is typically lower because of less emission by clouds passes to the surface through the atmosphere below the clouds."

**30) Lines 312-313: In addition, the summer cloud often radiate LW as a black-body, so an increase in LWP/IWP will not much affect their LW radiation in summer.**

✓ Thank you for pointing to this aspect. We included the discussion on this aspect in the previous paragraph related to the moist anomaly period where we also discussed about the influence of water vapor in presence of optically thick clouds. The following sentences were added and rephrased in the manuscript in the previous paragraph: "In contrast to other seasons, in summer the mean LW CRE during moist anomalies is not higher than under normal conditions due to several factors. First, cloud occurrence does not change much between normal and moist conditions in summer. Second, in summer clouds often emit LW radiation as black bodies and therefore, an increase in LWP and/or IWP does not essentially affect their LW radiation. Moreover, a similar LW CRE in summer between moist and normal conditions may be caused by influence of water vapor in presence of optically thick clouds as was previously described by Cox et al. (2015)."

**31) Lines 328-330: Do the author say that if LWP and IWP and frequency of occurrence of clouds do not vary, the cloud properties cannot vary? What about droplet size and aerosols affecting the SW CRE?**

✓ SW CRE estimations used in the current manuscript were based on the RRTMG model (Ebell et al., 2020) output and therefore there is no interaction between aerosols and cloud properties included.

✓ In the RRTMG model, Ebell et al. (2020) use a constant climatological profile of aerosol properties. The variability of the aerosol load is currently not available and thus it is one of the sources of uncertainties of the modeled CRE. Since variability of the aerosols is not included into the used CRE, the changes in CRE which are discussed in this sentence are not associated to aerosol properties.

✓ At SW the extinction by cloud droplets is defined by the scattering. The scattering efficiency at SW is constant (about 2) since it is practically in the optical limit here. Extinction is determined by the optical depth which is directly proportional to LWP and proportional to $1/r_{eff}$, where $r_{eff}$ is the effective radius of droplets. For a constant LWP, the projection area of cloud particles is inversely proportional to the effective radius of droplets. Thus, a change by a factor of 6 would require a 6-fold change or the effective radius of droplets. According to a number of studies focused on microphysical retrievals in the Arctic (e.g. Turner 2004, Shupe et al. 2005) cloud droplets in the Arctic are typically 5-8 micrometers in radius. Therefore, the variability in the cloud droplet sizes is not likely to be the major contributor to the factor of 6 variability in SW CRE.

**32) Lines 344-347: Omit or shorten.**

✓ Since some question related to the uncertainty sampling arose and were highlighted by the reviewer 2, we decided to keep this part as it is. Because it explains that the defined anomaly types were uniformly distributed and their occurrence were not associated with a dominant time period.

**33) Line 408: Are the percentile threshold values taken here from the Microwave radiometer data or radiosounding data?**

✓ Thank you for highlighting this point. The percentile threshold values are taken from the microwave radiometer data. We clarified this in the following sentence: "Therefore, we estimated the IWV value for each radiosonde profile and classified the profile using the thresholds defined in Sec. 1 based on MWR data."

**34) Lines 422-426: remove from here, because this part belongs to " Summary and conclusions".**

✓ Removed

**35) Line 440: Was the correlation analyzed? If not, do not use the word "correlate".**

✓ The word "correlated" was replaced by the word "associated".

**36) Lines 442-443: Again, what is specifically meant by "air circulating in the Arctic"?**

✓ Rephrased: "Dry periods in Ny–Ålesund are mainly related to air masses originating from latitudes north of the Arctic Circle, which is consistent with previous studies (Maturilli and Kayser, 2017a; Dahlke and Maturilli, 2017; Wu, 2017; Mewes and Jacobi, 2019)."

**37) Line 445: Statistically significant?**

✓ We replaced "significant" to "large".

**38) Line 446: "counterclockwise air circulations" could be "cyclonic".**

✓ We rephrased the following sentence: "In winter and spring dry conditions are associated with a low pressure system over the Barents sea causing northerly flow to the Svalbard region.".

**39) Line 447: What is meant by "this type of circulation"?**

✓ Rephrased: "In winter and spring dry conditions are associated with a low pressure system over the Barents sea causing northerly flow to the Svalbard region. This is in agreement with the results from Mewes and Jacobi (2019), who showed that in winter this northerly air mass transport to Svalbard region is related to the North Pacific pathway, which causes cold anomalies for the Svalbard region."

✓ Please note that this part was shifted in the revised version of the manuscript since we reorganized the Section "Summary and conclusions". Now it is placed in results for "-IWV" anomaly period.

**40) Line 449: See my earlier comment about these directions. I could not see these results in the figure.**

✓ In order to cut the length of the manuscript and since the anomaly periods "-T-IWV" and "+T+IWV" are a particular case of "-IWV" and "+IWV" periods, we decided to exclude

the  discussions related to "-T-IWV" and "+T+IWV" anomalies from the manuscript. We removed these sentences related to findings referred to "-T -IWV" anomaly type.

41) Lines 444-450: This part is far too long and the dynamical part is not well enough explained.

✓ We made this paragraph shorter by removing the part related to the "-T-IWV" anomaly type. Please see the answer to the previous comment.

42) Figure 8: From which level (height) the radiosonde IWV is taken?

✓ Calculations of IWV were based on the temperature, air pressure and humidity profiles from radiosondes. As radiosonde humidity sensor measurements are affected by low temperatures, their data reliability in the upper troposphere is limited particularly in the Arctic. Thus, we calculated IWV over a column from the surface up to 8 km height. In general, the largest contribution to the column atmospheric water vapor originates from the lowest altitudes up to 4 km (Nomokonova et al, 2019, Maturilli et al., 2016). Since the values of absolute humidity are quite low at altitudes higher than 8 km, we do not expect a substantial change in IWV values calculated for a column up to 8 km and up to higher altitudes of radiosonde profiles.

**Reply to reviewer #2**

We would like to thank the reviewer #2 for the constructive comments, which helped to improve the manuscript. We have considered all the recommendations.  In the following reply we repeat the statements of the reviewer (in red) and the reply to each statement of the reviewer (in black). Note that in the statements of the reviewer the line and figure numbers refer to the original manuscript and may have changed in the revised version.

**1 Specific comments:**

The main concern I have is associated with the sampling uncertainty; I speak more about this below.

1) Why wasn't the cloud radar included in the instrument list / description section? It is key for the CloudNet products, which are key for the rest of the paper.

✓ You are right that the Doppler cloud radar is a key instrument for Cloudnet and we agree on it. However, all the necessary information on all the instruments including also a Doppler cloud radar was provided in the previous study Nomokonova et al., (2019) that we also referenced to in the current manuscript at the beginning of the Section 2 (Instrumentation and data products).  In the current manuscript we paid more attention to describe the instruments and data products that have not been described and included in the previous study (Nomokonova et al., 2019). Therefore, we only briefly described instruments in

the current manuscript. If we include the detailed information on all the instrument it will make the manuscript too long and will lead to a repetition. And the first reviewer has asked us to shorten the existing manuscript. This is why we would like to keep the section as it is.

2) Why were NWP thermodynamic profiles used in the CloudNet classification, and not the profiles retrieved from the HATPRO?

   ✓ Please note that Cloudnet processing is a part of the ACTRIS European Research Infrastructure (www.actris.eu) focused on research of aerosols, clouds, and trace gases. Cloudnet either uses radiosonde and NWP data. NWP thermodynamic profiles are commonly used as input for Cloudnet by the community. In this study we use Cloudnet products as they are provided. We agree with the you that the microwave radiometer (MWR) HATPRO has certain advantages. One of the main advantages is continuous measurements with high temporal resolution. Nevertheless, this instrument cannot provide reliable observations during rain and conditions of condensation (fog) when the radome of the instrument is wet. Such cases are flagged and could not be used as input for Cloudnet. In addition, MWR do not provide the vertical resolution in temperature and humidity profiles needed for Cloudnet. In fact, the figures below show a comparison of temperature profiles retrieved from MWR HATPRO and NWP ICON model output for Ny-Ålesund with measured temperatures from radiosondes. MWR HATPRO shows a smaller bias in the lowest 1 km in comparison to NWP ICON model output. While the standard deviation of the temperature difference between MWR HATPRO and radiosondes is much higher in comparison to the ones between ICON model output and radiosondes for altitudes higher than 2 km. The standard deviation of the temperature difference between MWR HATPRO and radiosondes increases from around 1.5 °C at 1 km height to 5 °C at around 10 km.

[Figure]

Figure 1. Temperature difference between MWR HATPRO and radiosondes (left) and NWP ICON model and radiosondes (right). Blue and red lines show the bias and the standard deviation, respectively.

All this information is provided in Nomokonova et al., (2019) and the link is provided in the current manuscript.

3) Line 138: the RRTMG should be referenced, as it is a critical component to this study
   - ✓ We added the references to Mlawer et al., 1997 and Barker et al., 2003.

4) Are the uncertainties in the retrieved cloud properties (and in particular the phase classification) important to this study?
   - ✓ The uncertainties of cloud properties and particular phase classification are, for sure, important for this study and they were discussed in detail in the previous study Nomokonova et al., (2019) published earlier in the same special issue. In Nomokonova et al., (2019) a detailed information on instruments, Cloudnet uncertainties and description of the method for cloud classification used in this study are provided. For classification Cloudnet algorithm identifies the 0 C° isotherm based on the wet-bulb temperature from the model data. Therefore, the model temperature uncertainty (Figure 1 here) may lead to a misclassification of liquid and ice. In addition, ceilometer signal can be attenuated in the first liquid layer and therefore, cloud particles above the first liquid layer might not be detected. These cases occur in multi-layer clouds and single-layer mixed-phase clouds. Since the calculations of some microphysical cloud properties are based on LWP retrieved by MWR HATPRO and therefore, might be also influenced by the accuracy LWP. IWC was calculated using method based on Z-T relationship (where T is a temperature and Z is radar reflectivity) described by Horgan et al., (2006). IWC has bias error and typical random error of 0.923 and 1.76 dB, respectively. The uncertainty of IWC retrieval estimated by Horgan et al, (2006) depends on temperature: -50 to +100% for T below -40 °C and ranging from -33 to 50% for temperatures above -20 °C (root mean squared errors are given with respect to the reference IWC). The uncertainty in the radar reflectivity also affects IWC retrieval. The total uncertainty of 2 dB correspond to uncertainty in IWC ranging from +40 to -30%. More information on uncertainties are provided in Nomokonova et al, (2019). All these uncertainties result in differences between the simulated and measured radiation fluxes estimated and discussed in Ebell et al. (2020), the latter is also provided in subsection 2.5 in the initial version of the manuscript. The resulting uncertainties in SW and LW CRE are similar to the ones found in other state-of-the art studies for other Arctic sites.

5) What vertical separation is required to identify multi-layer clouds?
   - ✓ The method for cloud classification was provided in previous study Nomokonova et al., (2019) and in current manuscript the reference to this method is mentioned in current manuscript. A cloud layer is defined as a layer of at least three consecutive cloudy height bins (~20 m each). We considered cases as multilayer clouds if two or more cloud layers were separated by one or more clear sky height bins (~20 m resolution).

6) Are the backtrajectories allowed to touch the surface? Or if they touch the surface, do you call that the origin point for the trajectory?

  ✓ The backward trajectories used in this analysis were limited only by the travel time going back up to 6 days.

  ✓ The calculations of the backward trajectories are based on gridded meteorological fields of NWP model of ECMWF. The lowest model levels closest to the ground follow the topography (Stohl et al., 1998). All the files of calculated back trajectories have the orography height ($Z_{oro}$) which correspond to the height above ground. We have checked the minimum height above ground for all the backward trajectories shown in Figure 2 in the manuscript they have never touched the surface.

  ✓ The main purpose of the use of back trajectories in this study was to show the direction of air masses towards Ny-Ålesund rather than vertical displacement of the air mases which is particularly important, for instance, for aerosol studies. Therefore, we think that for a cluster analysis of backward trajectories that allows to discriminate distinct flows of two different anomaly types ("-IWV" and "+IWV") provided in this study with the emphases on the direction of air mases the information obtained from backward trajectories is still relevant.

7) Line 168: The HATPRO period is from 2011 to 2018. This isn't a very long period. Using the longer radiosonde record, how representative is the 2011-2018 period? It is clear that the mean values won't be the same (hence the trends over time), but is the range of variability (e.g., standard deviation of the IWV)) the same for the short period as the longer period? Ditto for the 2016-2018 period.

  ✓ We have checked the representativeness of the derived monthly values of IWV from MWR HATPRO and radiosondes for different periods. Figure 2 below shows the 10th, 50th, and 90th percentiles of monthly IWV values derived from different instruments. The monthly values of IWV derived from radiosondes for the periods from 1993-2018 and from 2011-2018 were compared with ones derived from MWR HATPRO. The results show a good agreement between IWV derived from radiosondes for both periods and from MWR HATPRO, the period from 2011-2018, which is used to identify the anomaly periods in the current manuscript. The monthly IWV for the longer period (1993-2018) is closer to the one from MWR HATPRO period with root-mean squared difference (RMSD) of the 10th and 90th percentiles of 0.6 and 0.5 kg/m$^{-2}$, respectively. Slightly higher monthly values of IWV from radiosondes were found for the same period as for MWR HATPRO (2011-2018) with RMSD of 0.4 and 1.2 kg/m$^{-2}$ for 10th and 90th percentiles, respectively.

✓ The shorter period from 2016-2018 is shown on the right figure below. The results revealed that percentiles of IWV from radiosondes (2016-2018) and MWR HATPRO (2011-2018) consistent with RMSD of 0.5 and 1.1 kg/m$^{-2}$ for 10$^{th}$ and 90$^{th}$ percentiles, respectively.

[Figure]

[Figure]

Figure 2. Monthly percentiles of IWV from MWR HATPRO and radiosondes at Ny-Ålesund for different periods. Black lines show 10th percentile (solid line), 50th percentile (dashed line), and 90th percentile (dotted line) of IWV calculated from radiosondes for the period from 1993 to 2018. Red lines are related to the correspondent percentiles of IWV retrieved from MWR HATPRO for the period from 2011 to 2018 (the same as shown in the current manuscript, Figure 1). Blue lines referred to correspondent percentiles of IWV derived from radiosondes for the period from 2011 to 2018.

✓ We added the upper figure to the supplement materials of the manuscript (please see the attached file of the Supplement material, Figure S3). We added the following sentence in the manuscript: "The representativeness and variability of IWV values obtained from MWR for the reference period from 2011 to 2018 was checked with respect to the long-term radiosonde record. The monthly values of IWV derived from radiosondes for the periods from 1993 to 2018 and from 2011 to 2018 were compared with ones derived from MWR HATPRO. The results are in a good agreement and summarized in Fig. S3 in supplement material. The monthly IWV for the longer period (1993-2018) is closer to the one from MWR HATPRO period with root-mean squared difference (RMSD) of the 10th and 90th percentiles around 0.6 and 0.5 kg m$^{-2}$, respectively. Slightly higher monthly values of IWV from radiosondes were found for the same period as for MWR HATPRO (2011-2018) with RMSD of 0.4 and 1.2 kg m$^{-2}$ for 10th and 90th percentiles, respectively."

8) I have a lot of questions regarding the sampling uncertainty, especially in the 2016-2018 dataset. The authors themselves hinted at this at line 225 when they say "Such a short period of time would probably not be representative: : :". Table 1 shows that the number of cases in these "outlier" categories is small (less than 100, often less than 50). This is, by far, the biggest weakness of the paper. The authors must augment their discussion to talk about sampling uncertainties, which includes the two questions above regarding representativeness.

✓ We agree that the short period of temperature anomalies might be not representative particularly for the "-T-IWV" event in autumn because it was a unique case which lasted continuously only for 5-days. That is why we decided to exclude the categories of temperature anomaly periods from the manuscript. This also helps to keep the manuscript shorter as the first reviewer has requested. So far, the vertically resolved cloud observations are available at Ny-Ålesund for the period from 2016 -2018 and were analyzed in this study. According to the definitions of the anomalies, the anomalous periods represent 20% of all the cloud observations (10% dry and 10% moist).

✓ We added the following sentence at the end of the Subsection 3.1 "Identification of periods…" in the revised version of the manuscript: "Since the number of cases for temperature anomalies ("+T +IWV" and "-T -IWV") is relatively low these types of anomalies were not considered for the further analysis.".

✓ The categories "-T-IWV" and "+T+IWV" were removed from the table 1 and the expressions "(not shown)"were added to the following sentences: "Periods with "+T +IWV" anomaly take a major part (about 67%) of all moist anomalous cases (not shown). In contrast, occurrence of "-T -IWV" periods is only about 35% of all dry anomalies in all seasons except winter, when the occurrence is almost 90% (not shown)."

✓ Please note that the numbers at the top of the figures 3 and 4 are the numbers of the 6-hourly periods. Within each 6 hourly period there are at least 70% of 30 s profiles. For instance, "-IWV" and "+IWV" events in autumn with 53 and 102 cases of 6-hourly periods include 34654 and 69605 cloud profiles, respectively. Thus, the number of analyzed cloud profiles for each anomaly type is enough to get statistics on cloud properties and their radiative effect and investigate their relation to changes in IWV. The vertically resolved cloud observations at Ny-Ålesund will continue within the (AC)[3] project and more data will be available in the future for further analysis.

✓ We also checked the representativeness of IWV thresholds used to identify the anomalous periods during the 2016-2018 period with the longer time period

(1993-2018). And the results of this comparison do not show essential difference. Please see the answer to the previous comment.

- ✓ The problems of the sampling uncertainty of anomaly periods were also discussed in the manuscript (Page 11, lines 344-362, original version of the manuscript). In the initial version of the manuscript we mentioned that the anomaly periods were uniformly distributed over a day. We added the figure with distributions of anomalous and normal cases among 6-hourly time periods in supplemental material (Fig. S2). The following expression "(Fig.~S2, supplement material)" was added to the text of the manuscript to link to this figure.

9) Line 227: you are talking about the "-T -IWV" cases, and stating that the LWC and IWC are larger than in normal conditions. This is indeed counterintuitive. Perhaps the authors could look at the column RH value (computed as IWV / saturated_IWV, where the temperature profile is retrieved from the HATPRO) to see if there are any differences in this value? It would seem that the column RH in the "-T -IWV" case must be larger than the normal conditions for the cloud water paths to be larger: : :

- ✓ In order to make the manuscript shorten and due to the low number of cases for the temperature anomaly periods anomalies ("-T-IWV" and "+T+IWV") we decided to exclude them from the detailed analysis. However, to answer the questions from reviewers related to "-T-IWV" period since this period was particular interesting, we provided a detail analysis. Please see the answer above to the specific comment # 27) from reviewer 1.

10) Section around line 249, where the discussion focuses on ice clouds: I think that the authors should consider breaking the analysis into high ice clouds (e.g., cirrus) and "boundary layer" ice clouds. Generally speaking, I would not expect the former to have much of a dependence on IWV, whereas I can see how the BL ice clouds could depend on changes in IWV.

- ✓ Following the suggestion of the reviewer we performed an analysis of the altitudes of the ice-containing profiles. Ice-only profiles (blue color in Fig. 4) have the median cloud top height of ~3 km which corresponds to the median cloud top temperature of -31°C. It is well-known that clouds formed at temperatures warmer than -25°C are formed heterogeneously via liquid phase. Therefore, as expected, the mixed profiles have lower median cloud top height (~1.6 km) and are characterized by the median cloud top temperature of -6 °C in summer and -15 °C during other seasons.
- ✓ This analysis agrees well to the expectations of the reviewer, i.e. ice-only profiles (cirrus clouds) occurring at higher altitudes are less sensitive to changes in IWV. Mixed profiles (boundary layer clouds) appear at lower altitudes and depends more on changes in IWV.
- ✓ The following sentences have been added to the updated manuscript: "Profiles with both liquid and ice phases (green columns in Fig. 4) have the median cloud top height and temperature of 1.6 km and –15 °C (not shown), respectively. In the case of pure ice

profiles (blue columns in Fig. 4) the cloud top height and temperature are 3 km and –31 °C, respectively. Thus, ice-containing clouds occurring at higher altitudes are apparently less affected by IWV anomalies."

11) Line 266: To state that there is a 2x increase in LWP is not really clear enough. If the LWP is less than 10 g/m2, then a 2x increase is 20 g/m2 which is still close to the uncertainty in the HATPRO LWP retrievals. Would those retrievals have sensitivity to the atmospheric state, or in other words, is this 2x change in LWP an artifact of the retrieval?

- ✓ We agree with the reviewer that a comparison of numbers close to the uncertainties should be made carefully. In the line 266 of the initial manuscript the two fold increase in LWP was from 70 $g/m^2$ (the mean value indicated by the orange marker) during normal events to about 130 $g/m^2$ during -T-IWV conditions. Both numbers by far exceed the uncertainty of the retrieval.

- ✓ Sensitivity of the HATPRO LWP is around 1-2 $gm^{-2}$. The uncertainty of the 15 $g/m^2$ of the LWP retrieval is related to the long-term drifts which depend on atmospheric state (IWV and T) and TB stability. The figure 5 shows the values averaged over weeks and therefore, such a long averaging reduces the influence of this drifts and thus the uncertainty is lower.

- ✓ Please also note that all the results and conclusions made for -T-IWV and +T+IWV have been removed from the manuscript.

12) Line 269: again, where the ice clouds are located vertically may be important for this statement.

- ✓ In the comment 10 pure ice profiles and those containing both liquid and ice phase were compared in terms of cloud top height and temperature. There indeed the height could explain a lesser sensitivity of pure ice profiles to IWV anomalies. Here (line 269 of the initial manuscript) IWP changes are given for all ice-containing profiles (pure ice + liquid and ice in profile). Figure 4 shows that in summer a majority of the ice containing profiles have both liquid and ice in profile. In the comment 10 we wrote that these profiles are more sensitive to anomalies but these profiles have nearly the same cloud top height in different seasons. Therefore, the lower sensitivity of IWP to IWV anomalies in summer cannot be fully explained by the different altitudes of cloud tops.

13) Line 291: need to add units to the "0 to 85"
- ✓ Thank you for highlighting this point. We added the units.

14) Line 464: "excess and shortage" are odd words here. I think this phrase must be changed to be more clear

- ✓ We changed the structure of the Section 4 "Summary and conclusions" by splitting the main finding into "+IWV" and "-IWV" and this expression was rephrased.

15) Line 466: "reduction of LWP and IWP by an order of magnitude" seems to suggest both are decreased by a factor of 10, when I believe you only mean the IWP is changed by a factor of 10.

- ✓ Thank you for pointing to this aspect. This sentence was wrong.

- ✓ In order to shorten the section "Summary and conclusion" and present only the main results, this sentence and some others were excluded from this section.

16) Line 500: "patterns" is misspelled
- ✓ Corrected

17) Fig 6a: Is the autumn "-T -IWV" bar where it is due to that one 5-day period? I think the answer is yes, and this is a great example indicating that the sampling errors must be better discussed. And a note should be made in the caption here.

- ✓ We agree that this case "-T-IWV" was relatively short to draw conclusions. This event was unique, and we provided the detail analysis. Please see the answer to the comment # 27) from the reviewer #1. However, we decided to remove the temperature anomaly periods from the manuscript and not analyze them additionally.

- ✓ We removed anomaly periods referred to "-T-IWV" and "+T+IWV" from the manuscript since they are related to the short periods. Please see the answers above.

**18)** Fig 6c: it would be nice to have a horizontal line at CRE= 0.
- ✓ We added a horizontal line at CRE =0. Please see Figure 6c.

**Additional changes**

Note that the line numbers refer to the revised manuscript.

1) A year in the references Ebell et al., 2019 was changed from "2019" to "2020" throughout the manuscript. The information in the reference list was also changed.
2) **Figure 3 and Figure 4:** There were mistakes in numbers of anomaly cases mentioned at the top of the bars in Figures 3 and 4. The numbers of cases for dry (dry and cold) anomalies were mixed

up with ones for moist (moist and warm) cases. The figures 3 and 4 with the wrong numbers were replaced by new with correct numbers.

3) **Page 3 (updated manuscript):** We replaced "… in this study" to "… the following section".
4) **Page 3 (updated manuscript):** We replaced "… has been operating" to "… has been operated".
5) **Page 7 (updated manuscript):** We rephrased the sentence: "Clear-sky cases were not included in the statistics of LWP and IWP."
6) **Page 9 (updated manuscript**): We added the following sentence "First, cloud occurrence does not change much between normal and moist conditions in summer.".
7) **Page 9 (updated manuscript**): The following sentence was rephrased: "Moreover, the similar LW CRE in summer between moist and normal conditions may be caused by influence of water vapor in presence of optically thick clouds as was previously described by (Cox et al., 2015)."
8) **Page 9 (updated manuscript):** We added: "..., respectively." To the end of the sentence. "Thus, the strongest SW CRE can be found in summer. Under normal conditions in summer and spring the mean SW CRE is -115 and -19 W m$^{-2}$, respectively."
9) **Page 9 (updated manuscript):** We rephrased the sentence: "In summer the cloud properties during dry anomalies do not change as strongly with respect to normal conditions compared to other seasons, …".
10) **Subsection 3.2 and 3.3** were combined. Now the title for subsection 3.2 is "Cloud occurrence and phase". The numbers for the following subsections in Section 3 were shifted correspondingly.
11) **Page 10 (updated manuscript):** The expression "under dry conditions" was redundant and was removed (page 12, line 372 in original version of the manuscript).
12) **Page 10 (updated manuscript):** The following sentence was moved to the previous paragraph: "Since the effects of SZA are mitigated…".
13) **Page 10 (updated manuscript):** We added the reference to Table 2 and changed the numbers of mean normalized SW CRE in the text according the Table 2. The changes were made in following sentence: "Table 2 shows that under normal conditions …"
14) **Page 10 (updated manuscript):** the following sentences were rephrased: **"**If anomaly cases uniformly distributed over a season, we expect no difference in surface albedo between anomaly and normal conditions. Thus, similar relative changes in CRE$_{SW}$ and nCRE$_{SW}$ and near-zero change in the surface albedo indicate similar SZA conditions for anomaly and normal cases. If anomaly cases distributed not uniformly over a season the diversion would show that anomaly cases were sparsely distributed over a season."
15) **Page 11 (updated manuscript):** We removed the following sentence: "Influences of water vapor anomalies have been previously discussed in details.".
16) **Page 12 (updated manuscript):** The following sentence "Figure 8 shows changes in anomaly occurrences." was rephrased by: "Figure 8 shows the frequency of occurrence of moist and dry anomalies for each season for the time period from 1993 to 2018."
17) **Page 12 (updated manuscript):** We removed the following sentence: "About a half of the profiles in autumn and winter of 1993 corresponded to dry anomalies.".
18) **Page 12 (updated manuscript):** Added the word "moisture" to the following sentence: "The main focus is  on the impact of anomalous moisture conditions …".
19) We changed the title of the **Section 3** from "Identification of periods … " to "Definition of periods …"
20) In acknowledgments "Project-Number" was replaced by "Project-ID".

21) Some expressions and sentences were slightly changed throughout the manuscript.

[revised manuscript text omitted]
 | +8.8(–94) | +0.2(–60) | +0.45(+173) | –4.9(+52) | –0.4(+115) | –0.08(–30) | -9.39 | -0.31 | 0.26 |

---

## Author Response (AR2)

**Reply to referee #1**

We would like to thank the referee #1 for the suggested minor comments, which helped us to improve the manuscript. We have considered all the recommendations. Below, the referee's comments are in red. Our replies are given in black. Please note, that in the statements of the referee lines and figures refer to the original manuscript and may have changed in the revised version.

**1 Minor comments:**

- Page 2, Line 22: define the time scale of this increase. Is it a seasonal increase, or increase in long-term winter mean (=trend)?

This information is related to the long-term period from 1996 to 2016.

We made the following changes in the manuscript:

"Dahlke and Maturilli (2017) analyzed the air mass transport in Ny-Ålesund in winter seasons from 1996 to 2016. The authors showed an increasing air mass transport through the North Atlantic pathway and reducing flow from the north."

- Page 5, Line 30: The sentence "found FLEXTRA back trajectory files with corresponding reaching time" could be rephrased. Data files do not have to be mentioned, it is enough to tell about the matching time. Specify also what is meant by "reaching time".

We rephrased this sentence: "We identified all 6-hourly periods with an anomaly class from June 2016 to October 2018. For each 6-hourly period we found a FLEXTRA back trajectory with time when it reaches Ny-Ålesund corresponding to the beginning of the 6-hourly period."

- Page 6, Lines 1-2: what is meant by this: "6-day trajectories are sufficient to capture the air transportation to the Arctic."? Why do you need 6 days to capture air mass transport to the Arctic?

We considered that a short time period may not capture a large-scale air flow direction while a long time period likely has large uncertainties.

We removed this sentence.

- Page 6, line 6: what is meant by this: "Mewes and Jacobi (2019) have shown that a similar horizontal air mass transport happens when air from the North Pacific flows into the Arctic."? What is" a similar horizontal air mass transport", especially if it comes from a completely different region?

The reviewer is right. The sentence is misleading. Mewes and Jacobi (2019) specify a pattern which they call "North Pacific pathway" (please see Fig. 2c in there). During this situation the air mass may reach the Svalbard region either from the Greenland or from Russia. We removed the sentence.

- Figure 6: This caption must be wrong. (a) is LW and (b) is SW, right?

Corrected

- Page 8, line 9: I assume 6b shows the SW and not LW. If so, this figure number here is not correct.

Corrected.

- Page 12, lines 17-19: Please rewrite this sentence to make it clearer. A suggestion: "Air masses originating from the North Atlantic are typically associated with increased surface temperature and IWV in the Svalbard region, whereas those originating from North Pacific are linked to a decrease in surface temperature and IWV (Dahlke and Maturilli, 2017)."

Corrected

- Page 12, lines 20-26: Nice that the sea ice is now mentioned, but I would like to know whether there is a connection between the sea ice decline and IWV trend. Even if you do not quantitatively analyze this, you can estimate it in more detail based on existing literature. Is it likely that the sea ice loss is directly associated with the winter trend in moist anomalies?

We added the following paragraph:

"Another possible contributor to the trends in IWV is a change in moisture fluxes. Boisvert et al. (2015) estimated that the contribution of the moisture flux to IWV is on average about 10% in the Arctic region. The authors mention that the main driver of the changes in the moisture flux is the sea ice cover. Therefore, changes in the sea-ice coverage around the archipelago of Svalbard might impact the occurrence of moist and dry events. A reduction of sea-ice coverage leads to more evaporation and allows for an increased moisture transport at the location. Boisvert et al. (2015) compared changes in sea-ice coverage and vertical moisture fluxes over the Arctic ocean based on the Atmospheric Infrared Sounder (AIRS) for the period from 2003 to 2013. The authors found that areas to the North and East of Svalbard have exhibited a significant increase in moisture fluxes and IWV, which were consistent with the sea-ice decline. In contrast the area to the West of Svalbard (where Ny-Ålesund is located) has showed a decrease in the moisture flux rate despite the decrease in the sea-ice extent. At the same time the authors found that IWV has increased even though the moisture flux rate has decreased. Therefore, Boisvert et al. (2015) concluded that for this area the increase in IWV might be associated with the air transport from lower latitudes rather than evaporation. Beside the changes of the sea ice coverage around the Svalbard archipelago a reduction of the local fjord ice cover has an impact on the local climate (Isaksen et al., 2016; Dahlke et al., 2020) and therefore, might also lead to a change in occurrence of anomalous atmospheric conditions at Ny-Ålesund."

- Page 12, lines 20-26: You mentioned that low sea ice coverage in spring and summer can enhance evaporation. In spring this is certainly true, but in summer the evaporation is really small in the sea areas, while most of the evaporation occurs on land.

We agree with the reviewer and decided to remove this sentence.

- Page 12, lines 20-26: I would like to add that reduced sea ice coverage enhances evaporation, but also allows for more efficient transport of moist air masses (without loosing their moisture on the way). So, sea ice also affects the transported moisture. Maybe you could add something like "allows for an increased moisture transport at the location".

We added the sentence: "A reduction of sea-ice coverage leads to more evaporation and allows for an increased moisture transport at the location."

- Summary and conclusions: This section has improved since the first version. However, I recommend some further changes: This section consist of 11 short paragraphs (shortest of them having only two sentences). Either group them so that you have only one (fluently written) paragraph for moist anomalies and one for the dry ones (+ a paragraph in the beginning and 1-2 in the end), or organize your conclusions with numbers/bullet points. In this current format, it is rather difficult to get an clear overview of the nice and valuable results of this study!

We added numbers to conclusions to highlight the results.

**Reply to referee #2**

We would like to thank the referee #2 for the second proofreading of the manuscript and his useful comments which help to improve the manuscript.

**Additional changes**

We updated the reference Dahlke et al., 2020.

[revised manuscript text omitted]
 | +8.8(–94) | +0.2(–60) | +0.45(+173) | –4.9(+52) | –0.4(+115) | –0.08(–30) | -9.39 | -0.31 | 0.26 |